# High-sensitive nascent transcript sequencing reveals BRD4-specific control of widespread enhancer and target gene transcription

Annkatrin Bressin [1,2,5], Olga Jasnovidova[1,5], Mirjam Arnold[1,3,5], Elisabeth Altendorfer[1], Filip Trajkovski[1,3], Thomas A. Kratz[1,3], Joanna E. Handzlik[1], Denes Hnisz [4] & Andreas Mayer [1] ✉

Gene transcription by RNA polymerase II (Pol II) is under control of promoters and distal regulatory elements known as enhancers. Enhancers are themselves transcribed by Pol II correlating with their activity. How enhancer transcription is regulated and coordinated with transcription at target genes has remained unclear. Here, we developed a high-sensitive native elongating transcript sequencing approach, called HiS-NET-seq, to provide an extended high-resolution view on transcription, especially at lowly transcribed regions such as enhancers. HiS-NET-seq uncovers new transcribed enhancers in human cells. A multi-omics analysis shows that genome-wide enhancer transcription depends on the BET family protein BRD4. Specifically, BRD4 co-localizes to enhancer and promoter-proximal gene regions, and is required for elongation activation at enhancers and their genes. BRD4 keeps a set of enhancers and genes in proximity through long-range contacts. From these studies BRD4 emerges as a general regulator of enhancer transcription that may link transcription at enhancers and genes.

Metazoan genomes are pervasively transcribed by RNA polymerase II (Pol II)[1,2]. Pol II transcription is not restricted to genes but also occurs in extragenic regions[1,3]. A main source of extragenic transcription is antisense transcription that originates in the opposite direction of a gene giving rise to non-coding transcripts[4]. Although antisense transcription from divergent promoters is widespread in mammalian cells[5–9], it can also arise within genes[8–10] leading to complex transcriptional architectures. A complete picture of the Pol II transcriptional landscape, especially at non-coding parts of the genome, has not yet emerged.

Gene transcription is under the control of proximal promoter and distal control elements such as enhancers, and is usually divided into different main phases. Following transcription initiation, the nascent RNA is produced during the elongation phase before transcription terminates at the termination zone[11–13]. Early transcription elongation has emerged as a regulatory hub when Pol II pauses in the promoter-proximal region at the majority of genes in mammalian cells[14–16]. Enhancers are required for the activated transcription of their target genes[17] and are often located within the same local chromatin interaction domain, termed topologically associating domain (TAD)[18,19]. Due to the sometimes large distance between enhancers and their cognate targets, mechanisms exist that bring them into physical proximity[20–22]. Enhancers can be located between genes or within genes. The systematic identification of intragenic enhancers has been challenging due to the overlap with their host genes and our knowledge of these enhancers is incomplete.

Enhancers are transcribed by Pol II[23–27] and enhancer transcription has been linked with enhancer activity[28,29]. Enhancer transcription

[1]Otto-Warburg-Laboratory, Max Planck Institute for Molecular Genetics, 14195 Berlin, Germany. [2]Department of Mathematics and Computer Science, Freie Universität Berlin, 14195 Berlin, Germany. [3]Department of Biology, Chemistry, and Pharmacy, Freie Universität Berlin, 14195 Berlin, Germany. [4]Department of Genome Regulation, Max Planck Institute for Molecular Genetics, 14195 Berlin, Germany. [5]These authors contributed equally: Annkatrin Bressin, Olga Jasnovidova, Mirjam Arnold. ✉e-mail: mayer@molgen.mpg.de

shares many features with Pol II transcription at genes including the formation of a pre-initiation complex prior to transcription initiation[25,30,31] and Pol II pausing shortly after transcription initiation[32]. Similar to genes, enhancers are often divergently transcribed by Pol II[26,27,33,34]. The mechanisms of how enhancer transcription is regulated and coordinated with Pol II transcription at the corresponding target gene have remained unclear.

According to the current view, transcription factors transmit signals between enhancers and their target genes[35]. Despite our incomplete knowledge of these factors and the molecular mechanisms of enhancer-target gene communication, the BET family protein BRD4, which has been implicated in a range of human diseases and emerged as a therapeutic target[36–38], is of particular interest in this context since it co-localizes to both target gene and enhancer regions[39–44]. At target genes, BRD4 helps to activate transcription elongation in mammalian cells[45–48] whereas its function in enhancer transcription is less well understood. Although it has been suggested that BRD4 is involved in enhancer RNA synthesis, its specific implication is still unclear mainly because studies have used pan-BET inhibition which can't discriminate between individual BET proteins[40,49–51]. Furthermore, BRD4 is thought to participate in higher-order genome organization and enhancer-promoter interactions[52–56] although a recent study found that BET proteins are dispensable for enhancer-promoter contacts[57]. Direct BRD4-specific roles in enhancer function have remained elusive.

Here, we developed a high-sensitive NET-seq approach to provide an extended high-resolution view on the genome-wide density of engaged RNA polymerase, including lowly transcribed enhancers. HiS-NET-seq identified thousands of new Pol II transcribed putative enhancers also within active genes of human cells. Using integrative functional multi-omics, this study reveals a direct and general role of BRD4 in enhancer transcription. The findings indicate that BRD4 is required for elongation activation at enhancers, similar to its role at cognate genes, and for maintaining proximity between transcribed enhancers and a set of target genes. This work is in line with the hypothesis of coordinated Pol II transcription at enhancers and their associated genes by BRD4 and its selected interactors.

## Results

### A high-sensitive NET-seq approach for mammalian cells

Current methods to profile the genomic position of transcriptionally engaged Pol II with nucleotide precision, such as native elongating transcript sequencing (NET-seq)[9,58], often suffer from low sensitivity. To overcome this main limitation and to reveal the fine structure of the Pol II density also at lowly transcribed genomic locations, we developed a high-sensitive NET-seq approach, called HiS-NET-seq. This approach includes short optimized labeling (10 min) of nascent RNA with the nucleotide analog 4-thiouridine (4sU) prior to chromatin isolation (Fig. 1a and Supplementary Fig. 1a–1f). 4sU labeling of RNA has been successfully used for the analysis of transcript abundance[59] and transcription[13,47,60,61]. In HiS-NET-seq, labeled nascent chromatin-associated RNA is affinity-enriched and converted into a NET-seq library (Fig. 1a and Supplementary Fig. 1a[62,63]); The position of transcribing RNA polymerase is revealed with nucleotide and DNA strand resolution by sequencing the regions corresponding to the 3′-ends of the original nascent RNA. HiS-NET-seq also uses spike-ins which allows quantitative comparisons between conditions.

HiS-NET-seq replicates correlated well ($r = 0.99$) indicating the robustness of this approach (Fig. 1b and Supplementary Fig. 1g). In contrast to standard NET-seq, HiS-NET-seq obtained a substantial 9-fold increase of informative Pol II reads, mainly due to a strong reduction of chromatin-bound mature RNAs such as sn/snoRNAs (Fig. 1c). Furthermore, the complexity of HiS-NET-seq libraries was higher as compared to previous NET-seq libraries (Supplementary Fig. 1h) indicating that a broader spectrum of transcribing Pol II was captured. The high sensitivity resulted in an overall strong increase in

coverage (Fig. 1d, Supplementary Fig. 1i and Supplementary Fig. 1j) revealing the Pol II density at a significantly larger fraction of transcribed genes, enhancers and antisense transcription units (ATUs) (Fig. 1e and Supplementary Fig. 1k). HiS-NET-seq maps Pol II densities at 6,598 more genes, 1787 more enhancers and 4,999 more ATUs in human cells as compared to NET-seq (Fig. 1e and Supplementary Fig. 1k). A comparison of HiS-NET-seq with genome-wide methods that have been used to identify actively transcribed genes and enhancers is given in Supplementary Table 1.

HiS-NET-seq data strongly correlated with data obtained by the other single-nucleotide resolution Pol II profiling method precision nuclear run-on sequencing ($r = 0.85$; PRO-seq[64]) (Supplementary Fig. 2a). The fraction of informative reads was higher in HiS-NET-seq as compared to PRO-seq and comparable to qPRO-seq[65] (Supplementary Fig. 2b). We observed the main difference between HiS-NET-seq and PRO-seq data in the promoter-proximal region of genes. HiS-NET-seq captured significantly more transcriptionally engaged Pol II in the promoter-proximal region and more engaged Pol II closer to the TSS as compared to PRO-seq and its variants (Fig. 1f and Supplementary Fig. 2c). Consistently, the correlation of the promoter-proximal Pol II density was lowest between HiS-NET-seq and PRO-seq, and highest between HiS-NET-seq and conventional NET-seq (Supplementary Fig. 2d). Taken together, HiS-NET-seq is a robust high-resolution RNA polymerase profiling approach well suited for the analysis of lower transcribed genomic regions that captures more engaged Pol II in promoter-proximal gene regions as compared to other approaches.

### HiS-NET-seq uncovers new transcribed putative enhancers

With the high sensitivity for enhancer transcription detection (Fig. 1e), we reasoned that HiS-NET-seq could uncover new transcribed enhancers. We first focused on extragenic regions. This analysis revealed bi- and uni-directional Pol II transcription in extragenic locations that did not overlap with annotated FANTOM5[66] enhancers or genes in human K562 cells. 1870 and 3957 of extragenic bi- or uni-directional transcriptional sites contained the known enhancer chromatin marks H3K27ac and H3K4me1 suggesting that they are putative transcribed enhancers (Fig. 2a; Supplementary Data 1).

For the identification of transcribed intragenic enhancers, we made use of widespread antisense transcription at genes detected by HiS-NET-seq (Supplementary Fig. 1i and Supplementary Fig. 1k; Methods). We classified gene-associated Pol II antisense transcription into convergent antisense (CAT) and divergent antisense transcription (DAT) (Fig. 2b, c; Supplementary Data 2). We excluded ATUs associated with alternative host gene promoters or overlapping genes. An integrative analysis with data obtained by GRO-cap, an approach that detects transcription start sites (TSSs) at nucleotide and DNA-strand resolution[26] (Supplementary Table 1) revealed that TSSs are enriched at the 5′-end of ATUs (Fig. 2d). In most cases, the TSS upstream of a CAT site was accompanied by a second TSS downstream and on the opposite strand (Fig. 2d). This observation indicates bidirectional intragenic transcription that was not associated with the host gene promoter or overlapping genes (Fig. 2e). HiS-NET-seq also detected CATs for which a reliable GRO-cap signal was absent, further illustrating the high sensitivity of the approach (Supplementary Fig. 2e). 5634 (48%) CAT sites overlapped with H3K27ac and H3K4me1 chromatin marks (Figs. 2f and 2g; Supplementary Data 3). Furthermore, no considerable signal from total RNA-seq was detected at CAT sites, supporting the view that they represent putative enhancers rather than alternative promoters of the host gene (Fig. 2g). Overall, we identified 11,007 putative Pol II transcribed enhancers that did not overlap with annotated FANTOM5 enhancers in human K562 cells.

At genes, sense transcription was usually accompanied by divergent antisense transcription originating upstream and in the opposite direction of the genic TSS (Fig. 2b–d), which is in line with

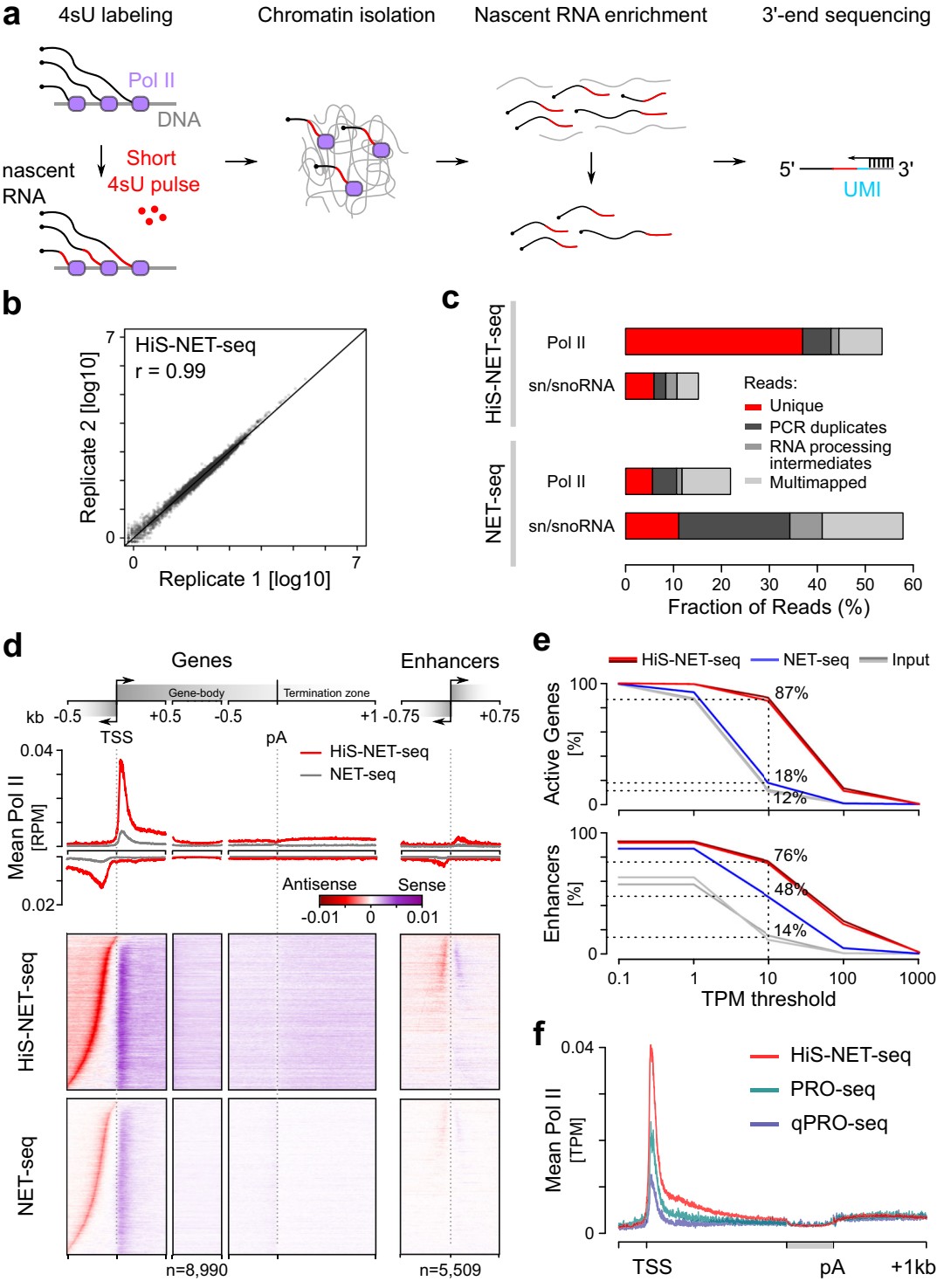

**Fig. 1 | HiS-NET-seq provides a more complete view on transcriptionally engaged RNA polymerase. a** Schematic view of the main steps of the HiS-NET-seq approach. The sequencing primer is indicated as black arrow (right panel). 4sU: 4-thiouridine (4sU, red); unique molecular identifier (UMI, blue). **b** Pearson's correlation analysis of Pol II occupancy per active gene for two biological replicate measurements of HiS-NET-seq ($r = 0.99$). Human gene counts were RLE-normalized (see Methods), and 0.5 pseudo counts were added. **c** Barplot shows the fraction of sequencing reads that mapped to Pol II transcribed regions (described in Methods) and sn/snoRNA genes. Unmapped sequencing reads and reads of masked regions are not shown. Fractions are indicated for HiS-NET-seq (top) and NET-seq (lower panel) data. **d** RPM normalized Pol II occupancy for individual nucleotides at indicated regions. Excluded were signal outliers above the 99.99- or 99.9-quantile

in HiS-NET-seq or NET-seq data, respectively. Pol II density at transcription start sites (TSSs) and polyadenylation (pA) sites was masked. TSSs and transcription directions are indicated by black arrows. Enhancers without Pol II signal in either HiS-NET-seq or NET-seq are not shown. kb: kilobase. **e** Quantification of Pol II occupancy measured by HiS-NET-seq and NET-seq, respectively. Element types as described in the Methods section include active genes ($n = 9454$) and FANTOM5 enhancers ($n = 6313$). Percent of transcribed elements with indicated TPM threshold or higher. **f** Pol II occupancy for individual nucleotides at gene regions ($n = 11,303$). Excluded were signal outliers above the 99.99-quantile in all datasets. The region from the TSS + 1.5 kb to the polyA site was scaled to 0.5 kb (gray). **b**–**d**, **f** Mean HiS-NET-seq values from two biological replicate measurements are shown. **b**–**f** Data were obtained for human K562 cells.

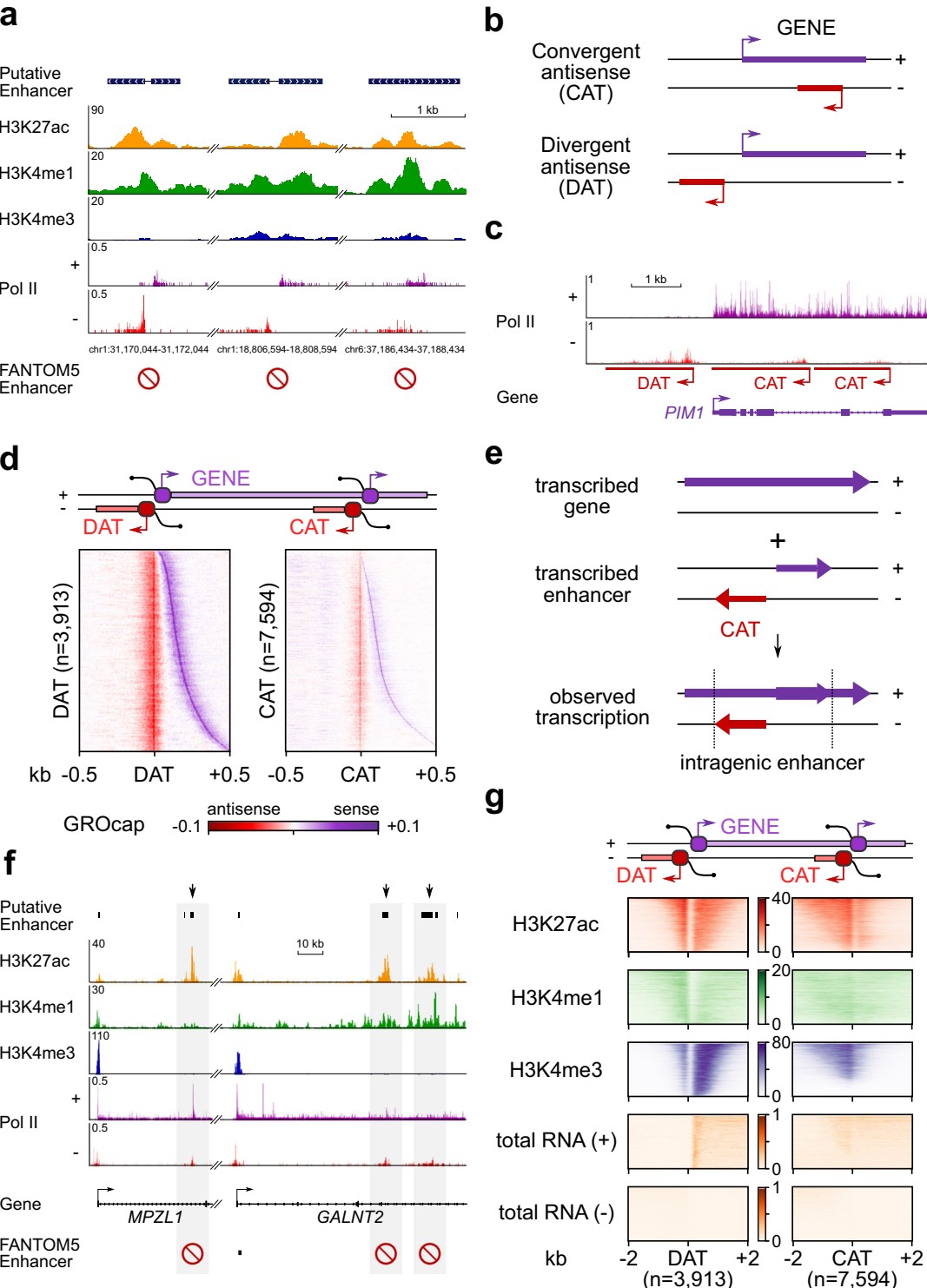

**Fig. 2 | Identification of new putative extra- and intragenic enhancers. a** Three putative extragenic enhancers with bi-directional Pol II transcription measured by HiS-NET-seq at indicated genomic loci. Elements overlap with enhancer marks such as H3K27ac and H3K4me1, but not with annotated K562 FANTOM5 enhancers symbolized by the red label. **b** Scheme of convergent (CAT) and divergent antisense transcription (DAT). Antisense and sense transcription are in red or purple, respectively. TSSs and transcription directions are indicated by arrows. **c** Gene track of transcriptionally engaged Pol II as determined by HiS-NET-seq at a representative gene. Same color code as in (**b**). **d** RPM normalized GRO-cap data[26] at identified CAT and DAT units overlapping with H3K27ac and H3K4me1 chromatin marks. The indicated center marks the identified 5′-end of CAT and DAT regions, respectively. Same color code as in (**b**, **c**). **e** Schematic representation of the

transcriptional complexity (third row) caused by the transcription of an intragenic enhancer (second row) within a transcribed host gene (first row). **f** Two gene examples with putative intragenic enhancers indicated by convergent antisense transcription (shaded regions). Elements overlap enhancer marks such as H3K27ac and H3K4me1, but not with FANTOM5 enhancers (red labels). **g** Heat map representations of signals of different histone marks (fold enrichment (FE) over matched input control; ChIP-seq) and of total RNA levels (raw; RNA-seq) at CAT and DAT units. Selected units overlapped with H3K27ac and H3K4me1 chromatin marks. **a**, **d**, **f**, **g** Chromatin marks were extracted from the ENCODE database[103]. **a**–**g** Data were obtained for human K562 cells. **a**, **c**, **f** HiS-NET-seq profiles are shown from two merged biological replicate measurements.

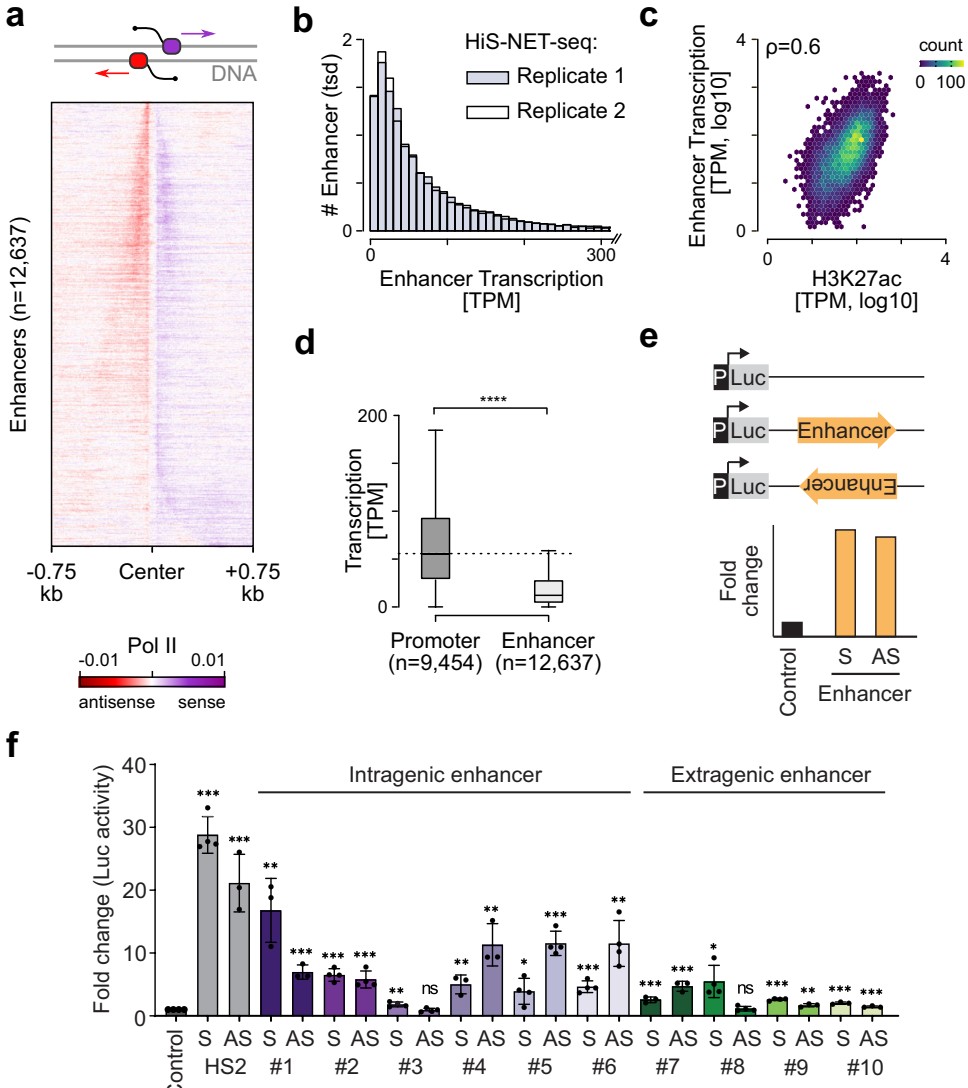

**Fig. 3 | Putative enhancer regions have characteristic enhancer features and activity. a** Heat map representation of RPM normalized Pol II occupancy measured by HiS-NET-seq for individual nucleotides at putative enhancer regions. Signal outliers above the 99.99-quantile were excluded. **b** Distribution of transcriptional activity at putative enhancers for two biological HiS-NET-seq replicate measurements. **c** Spearman's correlation analysis of Pol II occupancy per putative enhancer for HiS-NET-seq and ENCODE's H3K27ac data ([103]; $\rho$ = 0.6). **d** Quantification of TPM normalized Pol II occupancy measured by HiS-NET-seq at putative enhancer (n = 12,637, described in Methods) and promoter regions (n = 9454, two-sided Wilcoxon rank sum test; ****p < 2.2e−16). The boxplot shows the median values as the center. The box is defined by the first to the third interquartile range. The whiskers extend this interquartile range by a factor of 1.5, not exceeding the minimum or maximum values. Outlier measurements that exceed the whiskers are not shown. **e** Schematic view of the enhancer assay. The putative enhancer sequence was tested in sense (S) and antisense (AS) orientation for its ability to increase the expression of the Firefly luciferase (Luc) compared to the control. As a negative control, an early SV40 promoter (P) driven luciferase reporter was used. **f** Luciferase reporter assay measuring enhancer activity in K562 cells. Putative enhancer sequences were tested in sense (S)- and antisense (AS) directions. Luciferase (Luc) activity was normalized to that of a co-transfected Renilla luciferase and further normalized to the negative control. The HS2 minimal enhancer sequence[110] served as a positive control. Mean fold change +/− standard deviation was calculated and statistical significance was determined using a two-sided t-test (***p < 0.001, **p < 0.002, *p < 0.033, ns: 0.12, ns: not significant). All constructs were tested in biological triplicate, except control construct HS2 S, #2, #3, #5, #6, #8, and #9S were tested in biological quadruplicate. Source data are provided as a Source Data file. **a**–**f** Data were obtained for human K562 cells. **a**, **c**, **d** Mean HiS-NET-seq signals from two biological replicate measurements are shown.

previous observations[5–9]. DAT sites overlapped with H3K27ac, H3K4me1 and also with H3K4me3 marks (Fig. 2f, g). H3K4me3 was significantly enriched at DAT locations as compared to CAT sites, representing a main difference between both classes of antisense transcription (Supplementary Fig. 2f). Annotation files of identified antisense transcription units and putative enhancer regions are provided and can be directly loaded into the genome browser (Supplementary Data 1–3).

Together, HiS-NET-seq provides a more complete view on the enhancer transcription landscape and especially on enhancers within transcribed genes.

## Identified putative enhancers show classic enhancer features and activity

Since HiS-NET-seq not only detects transcribed enhancers but at the same time discloses the density of engaged Pol II at these sites, we next investigated the fine structure of transcribed putative enhancers. The nucleotide and DNA strand resolution of HiS-NET-seq clearly resolved Pol II transcriptional activities that originated within close proximity such as divergent transcription at the majority of enhancers (Fig. 3a), extending previous observations[26,27]. The signal intensity was similar for sense and antisense transcription at most enhancers. At a set of

enhancers (36%) the intensity was higher on one strand. When the signal at the opposite strand dropped below the detection threshold, enhancers appeared as uni-directionally transcribed (Fig. 3a and Supplementary Fig. 3). The signal intensity varied strongly between enhancers (Fig. 3b). Enhancer transcription correlated with H3K27ac (Fig. 3c), a histone mark that has been linked to enhancer activity[67–69]. The median HiS-NET-seq signal at enhancers was 4.6 times lower as compared to bidirectional transcription at gene promoters indicating that enhancers are generally less transcribed (Fig. 3d).

We next analyzed if these newly identified potential regulatory regions possess enhancer activity. We tested this for 10 selected putative extragenic and intragenic enhancers identified by HiS-NET-seq (Supplementary Data 4) using a dual reporter assay (Fig. 3e). All tested intragenic enhancers included CAT elements. Notably, out of the 10 putative regulatory regions, 8 led to a higher reporter gene expression irrespective of their orientation as compared to the negative control, indicating enhancer activity (Fig. 3f). The other two putative regulatory elements (Fig. 3f, #3 and #8) showed a significant increase in reporter gene expression only in one orientation and therefore it remained unclear whether they serve as potential enhancers or promoters. We conclude that HiS-NET-seq can reveal active enhancers opening a new avenue for the identification of intra- and extragenic enhancers in cells.

## BRD4 regulates enhancer transcription genome-wide

Despite the more complete view on enhancer transcription, the regulatory mechanisms remained unclear. To gain insights into the regulation of enhancer transcription, we focused on BRD4 because of its general implications in Pol II transcription and emerging role in enhancer-target gene communication[38,51]. In order to uncover potential direct functions of BRD4 in enhancer and target gene transcription, we first determined the genomic binding sites of BRD4 using chromatin immunoprecipitation with reference exogenous genome (ChIP-Rx)[47,70]. This analysis showed that BRD4 predominantly localized to putative enhancer and promoter-proximal gene regions (Fig. 4a and Supplementary Fig. 4a). The peak occupancy of BRD4 was 160 nt downstream of the TSS at genes and spanned a region +/− 100 nt from enhancer centers (Supplementary Fig. 4b). The binding intensity of BRD4 was similar at extragenic and intragenic enhancers (Fig. 4a, b). Interestingly, BRD4 also co-localized to the majority (61%) of putative enhancers (Supplementary Fig. 4c).

Integration of BRD4 ChIP-Rx and HiS-NET-seq data revealed that BRD4 occupancy correlated with enhancer transcription (Supplementary Fig. 4d), where enhancers with higher BRD4 binding showed significantly more enhancer transcription compared to enhancers with lower BRD4 levels (Fig. 4c). BRD4 was also enriched at CAT units (Supplementary Fig. 4e). Moreover, H3K27ac, H3K4me1 and H3K4me3 were enriched at genomic BRD4 binding sites (Supplementary Fig. 4f).

To further elucidate a direct and causal role of BRD4 in widespread enhancer transcription, we used a human cell line in which a degradation-tagged version of BRD4 is expressed from the endogenous locus[47]. The degradation tag (dTAG) allows rapid BRD4-selective degradation in cells upon exposure to the degrader (Fig. 4d). Degrader treatment led to an immediate reduction of BRD4 isoforms by more than 95% within <2 h (Fig. 4e). As an immediate consequence and consistent with our previous work[47], acute BRD4 ablation led to a strong decrease of nascent transcription at active genes (Fig. 4f). HiS-NET-seq captured a reduction in Pol II transcription at 6088 (2.7 times) more genes as compared to conventional NET-seq providing a more complete view of the effect (Fig. 4f). Strikingly, the higher sensitivity now also detected an immediate and strong reduction of transcription at extragenic and intragenic enhancers upon BRD4-specific degradation (Fig. 4g). Notably, the decrease in transcription was strongest at enhancers with a significant reduction of BRD4 binding after 2 h of

treatment, defined as BRD4-sensitive enhancers (Fig. 4h). Taken together, these studies suggest a direct role of BRD4 in the regulation of genome-wide enhancer transcription in human cells.

## BRD4 ablation synchronously attenuates elongation at enhancers and target genes

Given the immediate collapse of enhancer transcription upon BRD4-selective degradation, we analyzed potential consequences on target gene expression. A reduction of transcription at annotated target genes of affected FANTOM5 enhancers was clearly visible at individual genes including *MYC* (Fig. 5a). To investigate this on a global scale, we performed H3K27ac HiChIP[71] which measures 3D contact frequencies between different genomic loci that are associated, in this case, with H3K27ac. We used the identified 3D contacts to assign putative enhancers to target genes based on evidence that enhancers are in close 3D proximity to their cognate genes in the cell nucleus[18,19,22]. This analysis revealed that a collapse of enhancer transcription upon BRD4 loss was accompanied by an immediate reduction in Pol II transcription of the corresponding target gene (Fig. 5b). Interestingly, the reduction was stronger at target genes with two or more assigned BRD4-responsive enhancers (Fig. 5b).

How was target gene expression impaired upon BRD4 loss? Acute BRD4 loss led mainly to an accumulation of Pol II in the promoter-proximal region of genes accompanied by a collapse of productive elongation at gene-body regions (Fig. 5c and Supplementary Fig. 5a). Both findings are indicative for a disruption of Pol II pause release and were in line with our previous observations[47]. Notably, with HiS-NET-seq we now detected an elongation defect at 95% of actively transcribed genes as compared to 44% in the previous study, strongly extending recent observations (Fig. 5c).

To further increase the temporal resolution, we performed HiS-NET-seq at an earlier time point (40 min; Supplementary Fig. 5b) after degrader treatment at which the first robust reduction of cellular BRD4 levels was detected (Fig. 4e). Consistent with the 2 h time point, the shorter treatment provoked a strong reduction of enhancer transcription and also of productive elongation at genes (Fig. 5d). Interestingly, at this early time point, the impact was similarly strong at enhancers and genes (Supplementary Fig. 5c). Although this finding suggested a synchronous reduction of transcription at both sites, we cannot rule out the possibility of subtle differences in response times below the 40 min time point.

The high resolution of HiS-NET-seq further revealed that similar to genes, Pol II transcription was predominantly reduced at the enhancer distal region suggesting a collapse of elongation at enhancers (Fig. 5d, Supplementary Fig. 5a and Supplementary Fig. 5c). This effect increased with the time of dTAG7 exposure (Fig. 5d and Supplementary Fig. 5c). To gain additional insights into the temporal order of events, we next performed ChIP-Rx experiments of BRD4 at this earlier time point (40 min) after dTAG7 treatment (Supplementary Fig. 5d). We found that BRD4 occupancy was strongly reduced after 40 min at a large set of extragenic and intragenic enhancer, and promoter-proximal gene regions (Fig. 5e). The decrease of BRD4 occupancy was similar upon 40 min and 2 h of degrader treatment indicating a uniform and concurrent reduction of BRD4 at enhancers and genes (Fig. 5f). Together, these results suggest that acute BRD4 loss rapidly and synchronously impairs transcription elongation at both enhancers and genes.

## Acute BRD4 loss reshapes enhancer-target gene contacts

Since BRD4 binds both enhancers and promoter-proximal gene regions (Fig. 4a, b), we speculated that BRD4 may coordinate enhancer and target gene transcription through regulatory long-range DNA contacts. To test this, we performed H3K27ac HiChIP[71] upon acute BRD4 degradation (Supplementary Fig. 6a). H3K27ac levels were not altered during the short exposure with the degrader (Supplementary

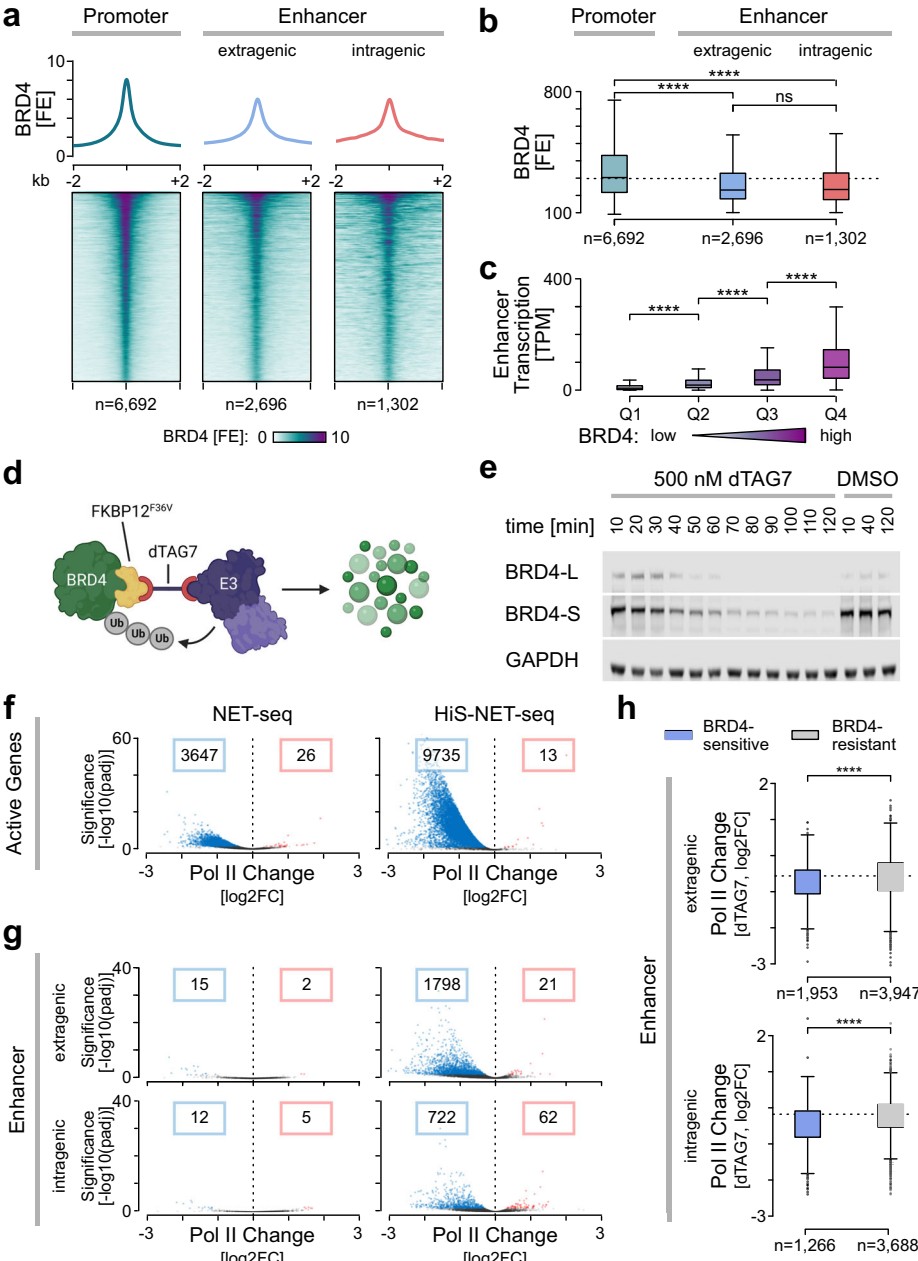

**Fig. 4 | BRD4 is required for genome-wide enhancer transcription.** Heatmap (**a**) and boxplot quantifications (**b**) of BRD4 occupancy (FE) at BRD4 peaks associated with promoter regions, extragenic-, and intragenic enhancers (one-sided Wilcoxon rank sum test; ****$p$ < 2.2e−16, ns: 0.72). ns: not significant. **c** Quantification of TPM normalized Pol II occupancy measured by HiS-NET-seq (two biological replicate measurements) at putative enhancer regions and FANTOM5 enhancers depending on their respective BRD4 levels. Boxplot shows four quantiles (Q1-Q4), from the lowest to the strongest BRD4 signal (one-sided Wilcoxon rank sum test; ****$p$ < 2.2e−16). **a–c** Mean BRD4 ChIP-Rx signals are shown for two biological replicate measurements. **b**, **c** See Fig. 3d legend for boxplot definition. **d** Scheme for BRD4-specific degradation using the PROTAC degrader dTAG7[120]. E2: ubiquitin-conjugating enzyme; E3: ubiquitin ligase; Ub: ubiquitin. Created with Bio-render.com. **e** Immunoblot for both BRD4 isoforms upon treatment with 500 nM dTAG7 and the DMSO control. An antibody against the HA tag, which was integrated along with the degron tag, was used. GAPDH served as a loading control. Experiment was performed in duplicate. Source data are provided as a Source Data file. **f–h** Pol II occupancy changes (log2) between 2 h of dTAG7 treatment and the DMSO control. The significance reports the FDR-adjusted $p$-values ($p$adj) from the Wald test calculated by DEseq2 as described in the Method section. The red and blue data points mark significant changes with an $p$adj smaller than 0.05. The number of genes or enhancers with decreased or increased Pol II occupancy are indicated within blue or red frames, respectively. Data is RLE normalized (see Methods) to spike-in controls from mouse NIH/3T3 cells. NET-seq data was re-analyzed from Arnold et al.[47]. Analyzed are active genes ($n$ = 10,789) (**f**), extragenic- ($n$ = 5900) and intragenic enhancers ($n$ = 4954) (**g**, **h**). **h** The Boxplot distinguishes between Pol II occupancy changes (log2; two-sided Wilcoxon rank sum test; ****$p$ < 2.2e−16) at putative enhancers with significant (blue, $p$adj <0.05) and non-significant (gray, $p$adj >0.05) reductions in DNA-bound BRD4 levels measured by ChIP-Rx upon 40 or 120 min of dTAG7 treatment. The $p$adj values are derived from Wald tests calculated by DiffBind as described in the Method section. The boxplot shows the median values as the center. The box is defined by the first to the third interquartile range. The whiskers extend this interquartile range by a factor of 1.5, not exceeding the minimum or maximum values. Outlier measurements that exceed the whiskers are depicted as individual dots. **a–h** Data were obtained for human K562 dTAG-BRD4 cells.

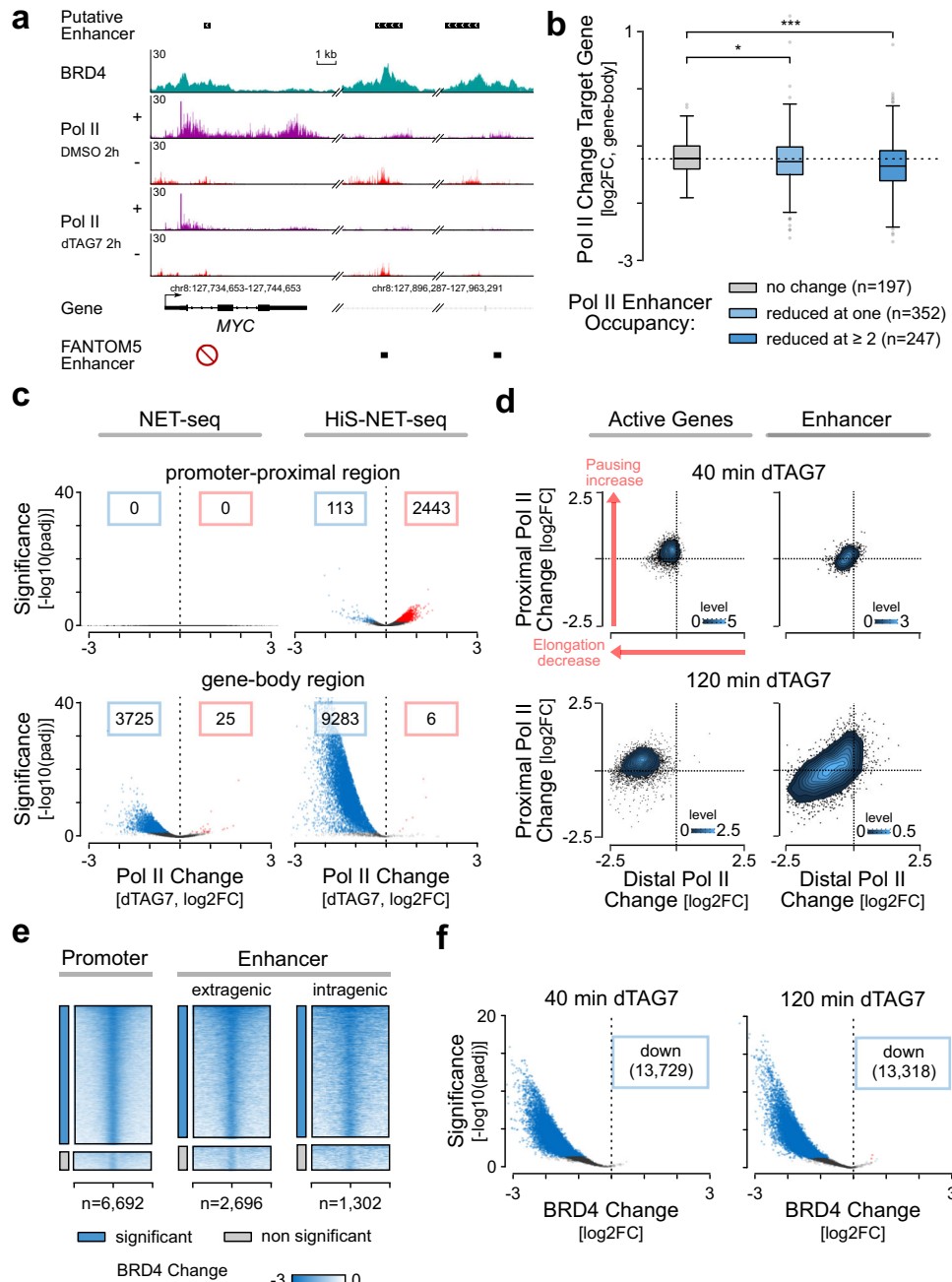

**Fig. 5 | BRD4 controls transcription elongation at genes and enhancers.**
**a** Target gene example with two associated FANTOM5 enhancers in K562 cells. All FANTOM5 enhancers of *MYC* are detected as putative enhancers by HiS-NET-seq and show significant Pol II occupancy reductions after 2 h of BRD4 degradation (*p*adj <7.8e−07). HiS-NET-seq profiles are shown from two merged biological replicate measurements. **b** Quantification of Pol II occupancy changes (log2) at gene-body regions of putative target genes. The Boxplot quantification distinguishes between the number of responsive enhancers (significant reduction in Pol II occupancy after 2 h dTAG7 treatment, *p*adj <0.05) connected to the corresponding target gene (one-sided Wilcoxon rank sum test; *p = 0.048; ***p = 2.1e−04). See Fig. 4h legend for boxplot definition. **c** Pol II occupancy changes (log2) between 2 h of dTAG7 treatment and the DMSO control. The significance reports the *p*adj values from the Wald test calculated by DEseq2 as described in the Method section. The red and blue data points mark significant (*p*adj <0.05) changes. Data is RLE normalized (see Methods) to spike-in controls from mouse NIH/3T3 cells. Analyzed are

promoter-proximal regions (*n* = 7324) and gene-body regions (*n* = 9730) of non-overlapping active genes. **d** Pausing matrices depict Pol II occupancy changes (log2) upon 40 min and 2 h of dTAG7 treatment in distal (*x*-axis) and proximal regions (*y*-axis) of active genes (40 min: *n* = 7090 and 2 h: *n* = 7324) and putative enhancer regions (40 min: *n* = 1017 and 2 h: *n* = 5422). Red arrows that are representatively shown in the first panel mark the figure quadrant associated with increased Pol II proximal pausing and decreased elongation at genes. **e**, **f** BRD4 ChIP-Rx occupancy changes (log2) at different regions. The significance reports the *p*adj values from the Wald test calculated by DiffBind as described in the Method section. **e** Heatmaps show significant (blue, *p*adj <0.05) and non-significant (gray, *p*adj >0.05) BRD4 occupancy changes at promoter, extragenic-, and intragenic enhancer regions after 2 h of dTAG7 treatment. **f** BRD4 occupancy changes at all detected BRD4 peaks after 40 min and 2 h of dTAG7 treatment (*n* = 16,233). The blue and red data points mark significant changes (*p*adj <0.05). **a**–**f** Data were obtained for human K562 dTAG-BRD4 cells.

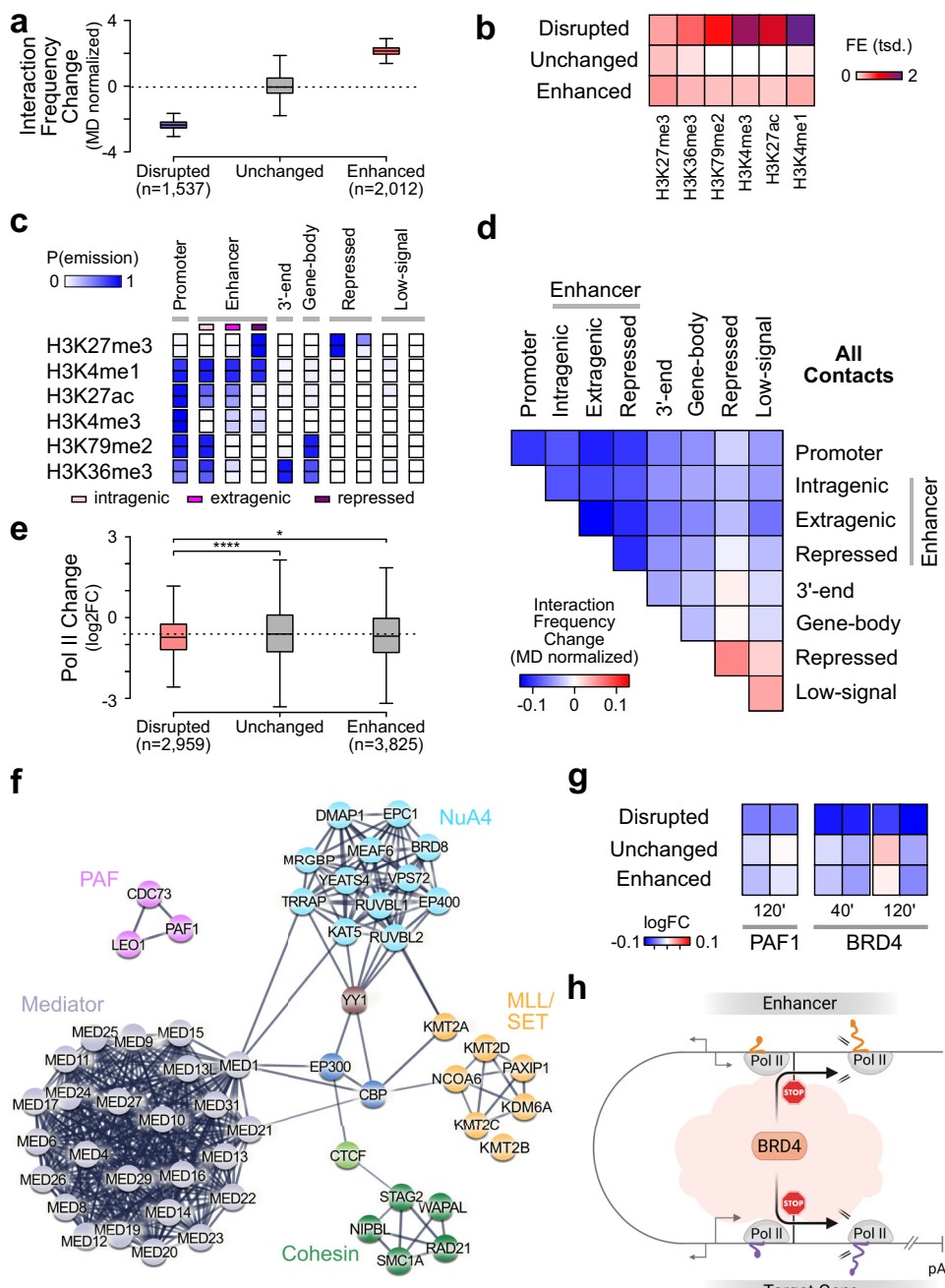

**Fig. 6 | Immediate change of regulatory contacts upon acute BRD4 loss.**
**a** HiChIP interaction frequency changes (log2) at reduced (blue), unchanged (gray, $n = 11,891,313$) and increased (red) 3D contacts upon 2 h of dTAG 7 treatment. See Fig. 3d legend for boxplot definition. **b** Heatmap shows mean occupancy (FE) of indicated chromatin marks at anchor regions from disrupted ($n = 2959$), unchanged ($n = 241,897$), and increased ($n = 3825$) contacts. **c** Depicted are state emission probabilities (P(emission)) of listed chromatin marks derived by chromHMM[72] from ten states in K562 cells. Annotations were manually assigned as described in the Methods section. **b, c** Chromatin marks were extracted from the ENCODE database[103]. **d** Mean interaction frequency changes (log2) at pairwise contacts between all indicated regions. **e** Pol II occupancy changes (log2) measured by HiS-NET-seq at anchor regions of disrupted ($n = 2959$), unchanged ($n = 241,897$) and increased ($n = 3825$) contacts (one-sided Wilcoxon rank sum test $*p = 0.03$;

$****p = 1.9e−10$). See Fig. 3d legend for boxplot definition. **f** Interactors of BRD4 involved in chromatin organization. Nodes represent significant interactors ($p < 0.05$, two-sided two-sample Student's $t$-test) identified by IP-MS. The edges correspond to "high confidence" physical interactions (STRING interaction score $>0.7$[121]). **g** Heatmap shows mean ChIP-Rx PAF1[47] and BRD4 occupancy changes (log2) upon 40 or 120 min of BRD4 degradation at anchor regions from disrupted ($n = 2959$), unchanged ($n = 241,897$), and increased ($n = 3825$) contacts. **h** Schematic model proposing the role of BRD4 in the coordination of enhancer and target gene transcription. Pol II pause release by BRD4 is indicated by black arrows. Transcription start sites are shown by gray arrows. DNA is depicted in gray. Nascent enhancer and genic RNA are shown in orange or purple, respectively. pA: poly-adenylation site. Created with Biorender.com. Data were obtained for human (**b, c**) K562 and (**a, d**–**g**) K562 dTAG-BRD4 cells.

Fig. 6b). We found that rapid BRD4-selective degradation led to an immediate change in the 3D interaction landscape. A set of contacts were significantly disrupted or enhanced upon BRD4 loss (Fig. 6a). Disrupted contacts showed a strong enrichment of chromatin marks associated with regulatory regions (Fig. 6b). To characterize the type of altered contacts in more detail, we classified the genome into the different main categories promoter, enhancer (intragenic, extragenic and repressed), 3′-end, gene-body, repressed, and low-signal, using common chromatin modification marker and chromHMM's[72] genome segmentation algorithm (Fig. 6c). Due to the limited spatial resolution of HiChIP data, a clear distinction between the promoter and promoter-proximal region was not possible and they were assigned to the same 'promoter' group. We found that contacts between enhancers and different regions of their target genes were immediately disrupted upon BRD4 ablation including contacts with the promoter, gene-body, and the 3′-end (Fig. 6d). We detected most disruptions for enhancer-promoter and enhancer-enhancer contacts (Fig. 6d and Supplementary Fig. 6c). We found similar disruptions of contacts for extragenic and intragenic enhancers (Fig. 6d). Interestingly, a stronger reduction of transcribing Pol II was detected at genomic regions with disrupted 3D contacts upon acute BRD4-specific loss (Fig. 6e).

In order to gain additional insights into the mechanism of how BRD4 maintains regulatory 3D genome contacts, we determined the BRD4 interactome in human cells. Native BRD4 immunoprecipitation coupled with mass-spectrometry (IP-MS) analysis identified known regulators of the 3D chromatin architecture including subunits of the Cohesin complex (RAD21, SMC1A, STAG2), CTCF, NIPBL, WAPL and YY1 as significant BRD4 interactors (Fig. 6f and Supplementary Data 5). Furthermore, BRD4 interacted with factors that have been implicated in enhancer-promoter communication, including histone acetyl transferases (EP300, CBP, the NuA4 complex), histone methyltransferases (MLL/SET family), the Mediator complex, and the PAF1 complex (PAF) (Fig. 6f and Supplementary Data 5). An integrative analysis with ChIP-Rx data revealed that at disrupted 3D contacts the BRD4 and PAF occupancy levels were most strongly reduced (Fig. 6g). These observations suggest that BRD4 can mediate long-range genome contacts through interactions with known factors involved in 3D chromatin regulation.

Together, the results are in line with the hypothesis that BRD4 can coordinate Pol II transcription elongation at enhancers and putative target genes likely through maintaining their proximity (Fig. 6h).

## Discussion

The increased sensitivity of HiS-NET-seq, which is mainly achieved by a two-step enrichment of nascent RNA combined with an efficient state-of-the-art NET-seq library preparation, provided a more complete view on transcriptionally engaged Pol II at genes and non-coding transcription units such as enhancers. This approach uncovered 11,007 new putative enhancers in K562 cells, mainly intragenic and lowly transcribed extragenic enhancers. Although intragenic enhancers have been observed and studied before[27,73,74], the identification of active enhancers within transcribed genes has been challenging since enhancer transcription overlaps with coding transcription and histone marks of the host gene. Here, we provide evidence that convergent antisense transcription (CAT) is indicative for transcribed intragenic enhancers and used this knowledge to systematically identify active enhancers within genes. CAT has been observed in previous studies[8–10,13,62,75,76], but its origin and function have remained unclear. In this study, we found that CAT to a large extent originates from putative intragenic enhancers suggesting a role in enhancer function. Although CAT may interfere with the expression of the host gene[73,77], more work will be required to clarify its role.

HiS-NET-seq data correlates well with PRO-seq but also reveals differences. The main difference was detected in the promoter-proximal region of genes where Pol II usually pauses before it is released into productive elongation. HiS-NET-seq captures significantly more engaged Pol II in this region as compared to PRO-seq methods. A potential explanation for this discrepancy could lie in the labeling of the nascent RNA. HiS-NET-seq uses 4sU labeling of nascent RNA under natural transcription elongation conditions in intact cells and likely also captures RNA polymerases that enter into a paused state or are recovering from a pause/arrest to resume RNA synthesis during the labeling time. Consistently, the correlation between HiS-NET-seq and conventional NET-seq data, which does not require labeling of nascent RNA, was highest especially in promoter-proximal regions of genes. Nuclear run-on based methods require nuclei isolation or cell permeabilization and a restart of transcription in the presence of biotinylated NTPs[78]. PRO-seq may miss a subset of paused/arrested RNA polymerases that may resist a restart of transcription[78,79].

Despite the increased sensitivity of HiS-NET-seq providing a more complete genomic occupancy profile of transcriptionally engaged Pol II, a current limitation is the relatively high amount of cells that are required as an input. This restricts the application of HiS-NET-seq to cell lines and primary cells at the moment that can be obtained in greater amounts. Since HiS-NET-seq uses 4sU labeling of nascent RNA the approach, similarly to other metabolic labeling-based RNA polymerase profiling methods, cannot be applied to tissues or whole multicellular organisms.

Enhancer transcription is regulated by BRD4. The BRD4-specific function in enhancer transcription has been unclear mainly because most prior functional studies have used pan-BET protein inhibitors or degraders disrupting all BET family proteins simultaneously. Here, we provide several lines of evidence that support a direct and general role of BRD4 in enhancer transcription. First, BRD4 binds to enhancers genome-wide. The binding of BRD4 to enhancers is in line with previous observations in other cellular contexts[39,41–44,46,80,81]. Second, BRD4 occupancy at enhancers correlates with enhancer transcription. Third, enhancer marks such as H3K27ac and H3K4me1 are enriched at BRD4 binding sites. Fourth, acute BRD4-specific ablation in cells causes an immediate collapse of enhancer transcription. Notably, the reduction of enhancer transcription was strongest at BRD4-sensitive enhancers. The high temporal resolution of inducible and selective degradation of transcriptional regulators[51,82,83], and HiS-NET-seq capturing the direct impact on Pol II transcription, before phenotypes are complicated by secondary effects, reveal a BRD4-specific function in enhancer transcription genome-wide.

BRD4 may link Pol II transcription at enhancers and their putative target genes (Fig. 6h). The following main observations support this view. First, BRD4 localizes to transcribed enhancers and promoter-proximal regions of active genes. Consistently, acute BRD4 loss leads to an immediate and similar reduction of BRD4 levels at enhancer and promoter-proximal regions. Second, BRD4-selective degradation provokes an instantaneous collapse of transcription elongation at enhancers and target genes before secondary effects start to accumulate. Third, BRD4 is required for enhancer-promoter proximity at a set of genes, likely through interactions of BRD4 with key factors that have been implicated in looping formation including the Cohesin complex, the Cohesin loading factor NIPBL, and the Mediator complex[42,52–54,84].

We speculate that a BRD4-mediated link between transcription at enhancers and their cognate genes can occur at the level of elongation at a set of genes. Our finding that BRD4 is required for Pol II pause release at enhancers and associated genes supports this hypothesis. Other BRD4 interactors are likely also involved including PAF. PAF is an integral component of the Pol II elongation complex that has been implicated in pause release[85–88] and enhancer function[89–91]. In this study, we found that BRD4 and PAF1 occupancy was most strongly reduced at genomic sites where 3D contacts were disrupted upon acute BRD4 ablation suggesting that BRD4 and PAF co-function in enhancer-target gene communication.

The proposed model on the coordination of enhancer and target gene transcription (Fig. 6h) is consistent with prior observations that enhancers associate with paused Pol II at target genes and with their implication in the regulation of transcription elongation[42,92–94]. Our view is also in line with a recently proposed condensate model for enhancer-promoter communication according to which hubs of transcription factors, coactivators including BRD4, and Pol II are formed to connect enhancers and target gene promoters[20,95]. Despite this evidence in support of the hypothesis that transcription at enhancers and a set of cognate genes is coordinated, more research will be required to clarify the links.

The multi-omics approach that we have used can now be applied to other factors to identify the players and their functions in the potential co-regulation of enhancer and target gene transcription to shed new light on enhancer-target gene communication.

## Methods

### Cell culture

K562 (ATCC, CCL-243) and K562 dTAG-BRD4[47] cells were cultured in RPMI 1640 (ThermoFischer Scientific, Cat.# 21875-091) containing 10% FBS Superior (Biochrom, Cat.# S0615), 5% penicillin-streptomycin (ThermoFischer Scientific, Cat.# 15070063). Cells were seeded at $5 \times 10^5$ cells/ml every two days. For Transfections, RPMI with 1x GlutaMAX (Cat.# 35050-038) was used. Cells were kept in culture for not longer than 4 weeks.

NIH/3T3 (ATCC, CRL-1658) cells were grown in DMEM (ThermoFischer Scientific, Cat.# 11995065) containing 10% FBS (Bovine Calf Serum Iron-Fortified, Sigma), 5% penicillin-streptomycin (ThermoFischer Scientific, Cat.# 15070063). NIH/3T3 were diluted to $2 \times 10^6$ cells/T75 flask every two days.

All cell lines were routinely tested for the presence of mycoplasma.

### HiS-NET-seq approach

**In vivo 4sU incorporation in suspension cells.** 48 h before an experiment, K562 cells were seeded at $5 \times 10^5$ cells/ml density in RPMI. Labeling of K562 was performed at a final concentration of 500 μM 4sU (Glentham Life Sciences, Cat.# GN6085) and a cell density of $1 \times 10^6$ cells/ml ($1 \times 10^8$ cells total). For labeling, cells were exposed to 0.5 M 4sU for 10 min at 37 °C and 5% $CO_2$. Cells were then placed on ice and fractionated according to the rapid fractionation procedure as described below.

### Degrader treatment combined with 4sU incorporation

Forty-eight hours prior to an experiment, K562 dTAG-BRD4 cells were seeded at $5 \times 10^5$ cells/ml in RPMI. For BRD4 depletion or matching DMSO control experiments $1 \times 10^8$ of K562 dTAG-BRD4 cells were collected and reconstituted at $1 \times 10^6$ cells/ml in fresh pre-warmed RPMI supplemented with 500 nM dTAG7 or an equivalent volume of DMSO (control). Degrader treatments were performed for 40 min or 2 h at 37 °C and 5% $CO_2$. For the final 10 min of the treatment, 4sU was added to a concentration of 500 μM. Next, cells were placed on ice and fractionated according to the rapid fractionation procedure. Four and two biological replicate measurements were performed for 40 min or 2 h of acute BRD4 degradation, respectively.

### Spike-in cell preparation

Twenty-four hours before the experiment $4 \times 10^6$ NIH/3T3 cells were seeded on fresh plates. The total number of cells used per experiment was ~$1 \times 10^8$. Cells were harvested and reconstituted to the $1 \times 10^6$ cells/ml in warm and fresh medium supplemented with 4sU (final concentration 500 μM) for 10 min at 37 °C. Cells were placed on ice and fractionated following the rapid fractionation procedure.

### Rapid fractionation procedure

Unless otherwise indicated, all procedures were performed on ice (4 °C) with pre-cooled buffers and under low light exposure. Cells were harvested for 2 min at $1150 \times g$. The pellet was gently resuspended in lysis buffer (PBS, 0.15% NP-40, 1× Protease inhibitor cOmplete, 50 U/ml SUPERaseIN, 25 μg/ml α-amanitin) and incubated for 2 min. Nuclei were collected at $500 \times g$ for 3 min. Nuclei were washed with cytoplasmic wash buffer (PBS, 0.1% Triton-X, 1 mM EDTA, 1× Protease inhibitor cOmplete, 50 U/ml SUPERaseIN, 25 μg/ml α-amanitin) and gently resuspended in 750 μl of glycerol nuclei buffer (20 mM Tris-HCl pH 8, 75 mM NaCl, 0.5 mM EDTA, 50% glycerol, 0.85 mM DTT, 1× Protease inhibitor cOmplete, 50 U/ml SUPERaseIN, 25 μg/ml α-amanitin). Chromatin was precipitated by rapid addition of 750 μl of nuclei lysis buffer (1% NP-40, 20 mM HEPES pH 7.5, 300 mM NaCl, 1 M urea, 0.2 mM EDTA, 1 mM DTT, 1× Protease inhibitor cOmplete, 50 U/ml SUPERaseIN, 25 μg/ml α-amanitin), and the sample was vortexed 5 times for 5 s, and then placed on ice for 2 min. Chromatin-associated RNA was collected at $18,000 \times g$ for 2 min. The pellet was washed twice with PBS and dissolved in 1.5 ml QIAzol Lysis Reagent per $5 \times 10^7$ cells (QIAGEN) supplemented with 100 μM DTT and 1 mM EDTA. To dissolve the pellet completely the sample was incubated at 40 °C for 1 h at 1000 rpm and homogenized using QIAshredder (QIAGEN, Cat.# 79654) for 1 min at $20,000 \times g$. The sample was stored at −80 °C.

### RNA extraction

Prior to RNA extraction 4sU-labeled spike-in RNA (in QIAzol) was added in a 1:8 ratio (NIH/3T3:K562). RNA was obtained by chloroform extraction using MaXtract High Density Phase-Lock-Gel tubes. RNA was obtained by isopropanol precipitation for 10 min on ice followed by centrifugation at $20,000 \times g$ and 4 °C for 20 min. The pellet was washed twice with 80% ice-cold ethanol, air-dried, and resuspended in nuclease-free $H_2O$. The samples were treated with TURBO DNase (Thermo Fisher Scientific) according to the manufacturer's instructions. The reaction was stopped by adding EDTA to the final concentration of 15 mM. RNA was purified using phenol:chloroform:isoamyl alcohol (25:24:1) (ROTH, Cat.# A156) and Phase Lock Gel Heavy tubes (QuantaBio Cat.# 733–2478) by centrifugation for 5 min at $12,000 \times g$. The RNA was collected by isopropanol precipitation, washed with 85% ethanol and resuspended in nuclease-free $H_2O$.

### Biotinylation and streptavidin pull-down of 4sU-RNA

100 μg RNA was biotinylated using MTSEA biotin-XX linker (Biotinum, Cat.# 90066) and purified by μMACS streptavidin MicroBeads (Miltenyi Biotec, 130-092-948) as described by Gregersen et al.[60]. with the following modifications. The μColumn (Miltenyi Biotec, 130-092-948) was washed three times with 1 ml of 65 °C pre-warmed pull-out wash buffer (100 mM Tris-HCl pH 7.5, 10 mM EDTA, 1 M NaCl, 0.1% TWEEN20 (v/v)), followed by three washes with 1 ml pull-out wash buffer at room temperature. The eluted RNA was purified using the ZYMO Clean & Concentrator-5 kit (ZYMO Research, Cat.# R1013).

### HiS-NET-seq and NET-seq library preparations

The HiS-NET-seq library preparation was conducted according to the nested-NET-seq protocol as described by Gajos et al.[62]. with the following modification. 2–3 μg of 4sU-labeled and biotinylated RNA was used as an input for the library preparation.

The conventional NET-seq library preparation were performed as previously described[63] with the following modification. The 'barcode DNA oligo' contained a random decamer sequence.

### Dot blot

The dot blot procedure was performed as described previously[60] with the following modifications. First, Hybond®-N+ hybridization membrane (GE Healthcare, Cat.# RPN303B) was used. Second, biotin was

probed by IRDye® 800CW Streptavidin antibody (LiCOR, Cat.# 925–32230) and visualized using a LI-COR Odyssey CLx imager.

## Immunoblotting

To obtain cytoplasmic, nucleoplasmic and chromatin samples, $1 \times 10^7$ cells were fractionated as described in the rapid fractionation procedure. The buffer volumes were downscaled according to the number of cells. For the dTAG7 time course experiment, whole cell lysates from $2 \times 10^6$ cells were prepared by incubation with 25–50 U benzonase and protease inhibitors (Roche, Cat.# 11873580001) in PBS on a shaker. For the H3K27ac analysis, subcellular fractionation was carried out using $1 \times 10^7$ cells as described in Mayer and Churchman[63]. For whole cell extract (WCE) $3 \times 10^6$ cells were harvested, washed in PBS and lysed in 300 µl RIPA buffer (Sigma) supplemented with 1× protease inhibitor cocktail (Roche) and 1 µl Benzonase (Millipore, Cat.# E1014-25KU). For histone 2B (H2B) probing, the membrane was stripped for 10 min in Restore PLUS Western Blot Stripping Buffer (Thermo Fisher, Cat.# 46430), washed and blocked prior to incubation with anti-H2B antibody (Santa Cruz Biotechnology, Cat.# sc-515808, 1:1000). The following primary and secondary antibodies were used: Pol II Ser2-P (3E10) (Active Motif, Cat.# 61083, 1:1000), Histone H2B (A-6) (Santa Cruz Biotechnology, Cat.# sc-515808, 1:1000), GAPDH (Ambion, Cat.# AM4300, 1:15000), HA tag (Cell Signaling Technology, Cat.# C29F4, 1:1000), H3K27ac (Abcam, ab4729, 1:1000), Tubulin (Abcam, ab18251, 1:3000), IRDye® 800CW Goat anti-Mouse IgG Secondary Antibody (LiCOR, Cat.# 926–32210, 1:15,000), IRDye® 800CW Goat anti-Rabbit IgG Secondary Antibody (LI-COR, Cat.# 925–32211, 1:15,000) and IRDye® 800CW Goat anti-Rat IgG Secondary Antibody (LiCOR, Cat.# 926–32219, 1:15,000). The signal was visualized using a LI-COR Odyssey CLx imager.

## Processing of NET-seq and HiS-NET-seq data

Data processing steps were applied as described by Gajos et al[62].. Briefly, the obtained sequencing reads were trimmed using cutadapt v3.4[96] (-a ATCTCGTATGCCGTCTTCTGCTTG -a AAAAAAAAAAAGGGG GGGGGGGGG -a GGGGGGGGGGGGGGGGGGGGGGGG -e 0.2 -q 5 --max-n 0.9) to remove sequenced fragments from the primers. Starcode v1.1[97] collapsed identical fragments (-d 0) with the same UMI sequence to one consensus read, removing PCR duplicate reads. The UMI decamer sequences were trimmed from the 5′ regions but the sequence information remained associated. The obtained sequencing read fragments were aligned to the human reference genome (GRCh38.p12)[97] using the STAR aligner v2.7.3a[98] (-clip3pAdapterSeq ATCTCGTATGCCGTCTTCTGCTTG -clip3pAdapterMMp 0.21 -clip3-pAfterAdapterNbases 1 -outFilterMultimapNmax 1 -out-SJfilterOverhangMin 3 1 1 1 -outSJfilterDistToOtherSJmin 0 0 0 0 -alignIntronMin 11 -alignEndsType EndToEnd). Next, a custom python script removed potential artifacts produced by mispriming of the RT primer if the UMI sequence corresponded to the genomic sequence adjacent to the aligned sequencing read. Furthermore, the custom script masked RNA processing intermediates produced during RNA splicing and 3′-end RNA cleavage. We excluded sequencing reads mapping to the 3′ most nucleotide position of annotated introns and exons, including the polyadenylation site. We masked the same nucleotide positions in the corresponding metagene visualizations to avoid artificial signal drops and potential misinterpretations of the occupancy profiles. Bedtools v2.29.2[99] was applied to mask regions[100–103], including transcribed regions of Pol I, Pol III, microRNA, miscellaneous RNA, rRNA, snoRNA, snRNA, tRNA, vault RNA, Y RNA, and blacklisted regions from ENCODE. Finally, the pipeline derives the Pol II occupancy tracks from the remaining uniquely mapped sequencing reads. We extracted the single-nucleotide 5′-positions from each sequencing read, corresponding to the 3′-end of the purified nascent RNA fragment.

## Processing of SI-NET-seq and HiS-NET-seq data with spike-in controls

The standard data processing pipeline for HiS-/NET-seq data, described in the previous paragraph, was applied to data with spiked-in control cells from the mouse (NIH/3T3) with the following adjustments. For mapping the sequencing reads to the reference genome, a joint reference from the human (GRCh38.p12) and mouse (GRCm38.p6) genomes was used. Furthermore, splicing and 3′-RNA processing intermediates from both species using the GENCODE v28 and M18 annotations[101] were removed. Next, RNA species from Pol I, Pol III, microRNA, miscellaneous RNA, rRNA, snoRNA, snRNA, tRNA, vault RNA, Y RNA, and blacklisted regions from ENCODE were masked in the human and mouse genomes. Finally, the sequencing reads mapping to the human genome were separated from those mapping to the mouse genome, resulting in a human and mouse Pol II occupancy data set for each sample. The Pol II occupancy in untreated mouse cells was used for data normalization.

## Comparison of HiS-NET-seq, standard NET-seq, PRO-seq, and qPRO-seq data

The systematic comparison of HiS-NET-seq, standard NET-seq, PRO-seq, and qPRO-seq data[65] was performed in two steps. First, all data sets were processed as described in detail in the 'Processing of NET-seq and HiS-NET-seq data' section with the following adjustments for PRO-seq and qPRO-seq. We changed the parameter setting of the following applications to accommodate for the paired-end sequencing mode of PRO- and qPRO-seq data: cutadapt, starcode, STAR, and custom scripts. We trimmed PRO-seq-specific adapter sequences (-a TGGAATTCTCGGGTGCCAAGGAACTCCAGTCAC -A GATCGTCG-GACTGTAGAACTCTGAACGTGTAGATCTCGGTGGTCGCCGTATCATT) and adjusted the UMI sequence length from 10 nt (HiS-NET-seq) to 6 nt at the forward and reverse strands. Second, we performed correlation and metagene analysis using DEseq2 v1.25.4[104] and deepTools2 v3.2.1[105], respectively. Data was normalized using the standard normalization strategies RLE[104] and RPM. Pol II density right at transcription start sites (TSSs) and polyadenylation (pA) sites were masked.

Next, we combined the described processing steps with additional data downsampling in the performance analysis. We applied seqtk v1.3-r106[106] to randomly select a subset of sequencing reads of equal size for each sample. The sample with the lowest sequencing depth served as a reference to estimate the new sample size of 14,556,851 sequencing reads. This step allows the direct comparison of informative reads between data sets obtained with the different methods.

## Identification of active genes and gene isoforms

This study defined cell line-specific active genes (K562 or NIH/3T3) as a subset from human v28 or mouse M18 GENCODE annotations[107]. A gene was classified as active using both the corresponding RNA-seq (ENCSR109IQO, ENCSR000OCLW[97]) and HiS-NET-seq data as follows. First, RSEM v1.3.1 113 quantified the number of transcripts produced by each gene and isoform in the respective single-end or paired-end mode using the RNA-seq data and the STAR v2.7.9a 105 alignment tool. Second, the genes with a TPM ≥ 1 were selected. Third, we refined GENCODE's annotation based on active gene isoforms by identifying the first and last active TSS and polyA sites. An active gene isoform contributed at least 10% to the overall gene activity. Fourth, genes without nascent transcription were removed. We used HOMER's v4.11.1 114 make-TagDirectory (-flip) and findPeaks functions (-style groseq -minBodySize 150,500 -tssSize 10) to annotate nascent transcription units using HiS-NET-seq data.

### Identification of active FANTOM5 enhancers in K562

Actively transcribed enhancer units were identified for the K562 cell line from annotated FANTOM5 enhancers. Data sets were extracted from the HACER database[107], which reported cell-type-specific FAN-TOM5 enhancer units and initiation sites identified by the NRSA application[108].

### Detection of CAT-, DAT-, and putative enhancer-regions

Convergent antisense transcription (CAT)-, divergent antisense transcription (DAT)-, and putative enhancer regions were derived from nascent transcripts of HiS-NET-seq data using HOMER v4.11.1[109], as described in the previous paragraph 'Identification of active genes and gene isoforms'. Antisense transcription units (ATUs), DAT and CAT units overlapped with annotated genes at the opposite strand relative to an actively transcribed gene. We distinguished between both types of antisense transcription units by their location, where DATs originate upstream (<1000 bp) and CATs downstream from the corresponding transcription start site of the gene. To eliminate the risk of spill-over effects from neighboring genes, we removed ATUs originating from overlapping genes. A transcript originates from an overlapping gene if either 50% or more overlap, or the transcription unit covers at least 90% of the gene. Finally, we removed transcription units overlapping the entire antisense gene region, as observed for many short genes.

Assigning CAT and DAT units is more complex if an actively transcribed gene expresses several transcript isoforms from multiple active TSSs. We defined active gene isoforms as described in the paragraph 'Identification of active genes and gene isoforms'. Some perceived CAT units originated from bidirectional transcription of downstream-located alternative TSSs. These sites were considered as DAT units. Putative intragenic enhancers were defined by the remaining CAT units that co-localize with the histone marks H3K27ac and H3K4me1 (ENCSR000AKP and ENCSR000EWC[103]).

Putative extragenic enhancers are defined by transcription units that are distal to active genes and have the histone marks H3K27ac and H3K4me1. We excluded transcription units originating from any GENCODE annotated promoter region, including genes that fail the activity threshold and are considered inactive. Furthermore, transcription units observed in the termination zone (polyA site +2 kb) of active genes were removed. We classified divergently transcribed units as bi-directional and unpaired units as uni-directional enhancers. To classify as a potential extragenic enhancer, bidirectional transcription needs to originate within <500 bp.

### Differential Pol II occupancy analysis

We tested for changes in the Pol II occupancy using DEseq2 v1.25.4[104] in different regions (Supplementary Fig. 5a), including active genes, promoter-proximal regions, gene-body regions, enhancer regions, proximal enhancer regions, and distal enhancer regions. After quantification, genomic regions with less than six sequencing reads across all samples were excluded. Next, we tested for significant changes between the conditions using DEseq2 (fitType=local). For data normalization, we applied the calculated scaling factors (relative-log-expression (RLE)) from the features in the mouse genome to the human observations, using the 'controlGenes' parameter. Significant changes showed an FDR-adjusted $p$-value ($p$adj) <0.05.

### Enhancer reporter assay

For the Dual-Luciferase Assay potential enhancer DNA sequences were PCR amplified from K562 cells and cloned in both sense and antisense direction into the pGL3-promoter vector by either conventional restriction enzyme cloning using BamHI-HF or SalI-HF (NEB, Cat.# R3136S and R3138S) or by Gibson Assembly (NEB, Cat.# E2611S). The primer sequences used for amplification are listed in Supplementary Data 6. As positive control, the minimal HS2 enhancer sequence was used[110].

K562 ($5 \times 10^5$) cells were transfected with 0.6 ng *Renilla* luciferase internal control plasmid (pRL-TK) together with the Firefly luciferase test plasmid (enhancer construct) in a molar ratio of 1:700. In addition, 4 μM electroporation enhancer (IDT) were added to each transfection reaction. If necessary, additional electroporation enhancer was added to ensure an equal amount of total DNA in all transfections. K562 cells were transfected using SF Cell Line 4D-Nucleofector™ X Kit S and Amaxa Nucleofector 4D. Cells were harvested 24 h post transfection and lysates were prepared using 100 μl 1× Passive Lysis Buffer (Promega, Cat.# E1910).

The reporter assay was performed using the Dual-Luciferase® Reporter Assay System (Promega) according to the manufacturer's protocol using 20 μL cell lysate. The luciferase activity was measured using the GloMax® Navigator Microplate Luminometer with dual injector (Promega) in a 96-well plate with following settings: 100 μL of the Firefly Luciferase Reagent (LAR II) was injected to each sample with a 2 s measurement delay time and measurement of luminescence with a 10 sec integration time. Subsequently, 100 μL of the Renilla Luciferase Reagent and Firefly quenching (Stop & Glo) was added with a 1 s measurement delay time, and measurement of luminescence with a 10 sec integration time. The data were processed using Excel and finally represented as the ratio of Firefly to Renilla luciferase activity. Statistics were performed with Graphpad Prism using a *t*-test.

### HiChIP

HiChIP was performed as described by Mumbach et al.[71]. with the following modifications. Per condition, three biological replicates were generated. $5 \times 10^6$ K562 dTAG-BRD4 cells[47] were treated with 500 nM dTAG7 or DMSO (control) for 2 h, followed by crosslinking with 1% methanol-free formaldehyde for 10 min at room temperature. The partially lysed nuclei were incubated with 375 U MboI (NEB, Cat.# R0147M) at 37 °C overnight. After biotin-dATP incorporation, chromatin shearing was conducted in a Covaris S220 sonicator for 6 min at intensity 4, duty cycle 5% and 200 cycles per burst. For IP, 5 μg of an H3K27ac-specific antibody (Abcam, ab4729) coupled to Pierce Protein A/G (Thermo Fisher Scientific, Cat.# 10004D) were used per sample. The eluted DNA was heated to 65 °C for 5 h, treated with 80 μg RNase A (Thermo Fisher Scientific, Cat.# 10753721) for 2 h at 37 °C and with 200 μg Proteinase K (Carl Roth, Cat.# 3719.2) for 45 min at 50 °C. 50 ng of the purified DNA was incubated with 2.5 μl of Tn5 (Illumina, Cat.# 20034197). NEBNext Q5 master mix (NEB, Cat.# M0544S) was used to amplify the library by 8 cycles. Lastly, the libraries were size-selected using AMPure XP beads (Backman Coulter, Cat.# A63881) at a ratio of 0.8 and subsequently 0.5 to enrich fragments of 200–750 bp and sequenced in PE75 or PE100 mode on an Illumina HiSeq 4000 or NovaSeq 6000 sequencer.

### Processing, normalization, and comparison of HiChIP data

For HiChIP data processing, we used the HiC-Pro v3.0.0[111] pipeline where raw interaction frequencies were extracted at a 5 and 10 kb resolution for the applied restriction enzyme (MboI: ^GATC) and the human reference genome (GRCh38.p12)[101]. For data normalization, we applied two alternative normalization approaches implemented by HiCcompare v1.8.0[112] and FitHiChIP v10.0[113].

For quantitative comparisons between conditions, we applied the global data normalization approach from HiCcompare[112]. The application removes batch and sequencing depth biases and accounts for the linear distance between interacting regions. Briefly, we performed the following steps. First, QDNAseq's v1.22.0[114] build-in get_CNV function identified copy number variations in the K562 dTAG-BRD4 cell line and excluded the respective regions. Second, low-complexity regions (blacklisted) annotated by ENCODE[103] were excluded. Third, we used the 'hic_loess' and 'hic_compare' functions to compute logarithmic interaction frequency changes, correct for systematic biases, calculate z-scores and p-values, and to perform multiple test corrections.

Interactions with an average expression <9 were excluded for comparison. Interactions with a minimum distance of 10 kb and a padj value < 0.05 were identified as significant changes between conditions.

## Identification of target genes from HiChIP data

For 3D contact identification, we used the FitHiChIP v10.0 application[113]. Significant 3D interactions were identified using pooled HiChIP data with 5 kb resolution from the control experiments. FitHiChIP's normalization method learns the relationship between coverage and genomic distance using a regression model. For this background model, we considered only interactions between H3K27ac genomic loci derived by ChIP-seq data from ENCODE[103]. 3D interactions that exceeded the expected coverage from the background model were considered significant, if the corresponding p-value was <0.01 after multiple test corrections. Target genes from putative enhancers were assigned by at least one significant 3D interaction.

## Segmentation and genome annotation

The human genome was segmented and annotated at a 10 kb resolution. We applied the chromHMM v1.19 [73]application to K562-specific chromatin marks from ENCODE data[103], including H3K27ac (ENCSR000AKP), H3K4me1 (ENCSR000EWC), H3K4me3 (ENCSR000EWA), H3K27me3 (ENCSR000EWB), H3K36me3 (ENCSR000AKR), and H3K79me2 (ENCSR000APD). First, the chromHMM application divided the human genome into 10 kb bins using the BinarizeBam function. Second, chromHMM's LearnModel function trained a multivariate Hidden Markov Model with ten states and reported the most likely states for each bin. In the last step, the resulting states were manually annotated into promoter (including the promoter-proximal region), enhancer (repressed, intra-, and extragenic), 3'-end, gene-body, repressed, and low-signal states. The promoter, intragenic-, and extragenic enhancer states showed high levels of H3K27ac. However, a unique chromatin feature of the promoter state was the high level of H3K4me3, which is absent or reduced in the three enhancer states. As described by[115], we assigned states with H3K79me2 to gene-body regions, whereas H3K36me3 was more abundant at the 3'-ends of genes (Supplementary Fig. 6d). Furthermore, we used H3K79me2 and H3K36me3 to distinguish intragenic from extragenic enhancers. Finally, we used the repressive H3K27me3 histone mark to define repressed genome regions and repressed enhancers[115]. States without significant enrichment of chromatin marks were annotated as low-signal states.

## ChIP-Rx

ChIP-Rx was performed as described by Arnold et al.[47]. with the following modifications. Per condition, two biological replicates were generated. $4 \times 10^7$ K562 dTAG-BRD4 cells were treated with 500 nM dTAG7 or DMSO for 40 min or 2 h and after formaldehyde crosslinking combined with $1 \times 10^7$ murine NIH/3T3 cells. For immunoprecipitation, 8 µg of a BRD4-specific antibody (Bethyl, Cat.# A301-985A50) were used. ChIP-Rx libraries were prepared using the NEBNext Ultra II DNA kit (NEB, Cat.# E7645S) according to the manufacturer's instructions. Libraries were purified using one volume of AMPure XP beads (Backman Coulter, Cat.# A63881) and size-selected (200–500 bp) from an 8% TBE gel (Thermo Fisher Scientific). Sequencing in PE100 mode was done on an Illumina NovaSeq 6000 sequencer.

## Processing of ChIP-Rx data

First, we used Bowtie2 v2.3.5.1[116] to align the sequencing reads to a joint reference which consisted of the human (GRCh38.p12) and mouse (GRCm38.p6) reference genomes using the paired-end mode with the parameter -k 1. We extracted the DNA-binding profiles (fold enrichment over matched input control (FE)) of BRD4 using MACS2's v2.2.7.1[117] bdgcmp function with the -m FE parameter. As recommended by the MACS2 developers, potential PCR duplicates were marked using PICARD's v2.24.2[118] markDuplicates function. Finally, the DNA-binding profiles of BRD4 corresponding to the human genome were separated from the BRD4 binding sites in the mouse genome.

## Differential binding site analysis from ChIP-Rx data

We used DiffBind v3.0.15[119] to identify differentially bound occupancy sites. First, DiffBind's 'dba.blacklist' function removed the signal from blacklisted regions of the human and mouse reference genomes (blacklist=DBA_BLACKLIST_HG38, blacklist=DBA_BLACKLIST_MM10). Second, peaks from all samples were summarized in consensus peaks and quantified (minOverlap=2, summits=300, bRemoveDuplicates=true, bSubControl=true). Third, DiffBind's 'dba.normalize' function adjusted the human consensus peaks based on the binned mouse genome (spikein=true, normalize=DBA_NORM_RLE, background=T). Finally, differential binding sites were identified using the DEseq2 package[104].

Furthermore, for the visualization of the occupancy changes (log2), we used the deepTools2's v3.2.1[105] bamCompare function with the parameters -p 20 --ignoreDuplicates --scaleFactorsMethod None. The previously calculated scaling factors from DiffBind were applied using the --scaleFactors parameter.

## Native immunoprecipitation with mass spectrometry analysis (IP-MS)

IP-MS experiments were performed as described by Arnold et al.[47]. Briefly, the K562 dTAG-BRD4 cells were lysed in a native IP buffer (20 mM Tris pH 8.0, 50 mM NaCl, 0.5% (vol/vol) NP-40, 10% (vol/vol) glycerol, protease inhibitor cocktail (Roche), phosphatase inhibitor cocktail (Roche) supplemented with benzonase (Millipore, Cat.# E1014-25KU). Nuclei were isolated by centrifugation and lysed in native IP buffer supplemented with benzonase (Millipore, Cat.# E1014-25KU). For IP 10 µg of the BRD4-specific antibody (Bethyl, Cat.# A301-985A50) or of isotype-matched IgG (Bethyl, Cat.# P120-101) and 1,5 mg Dynabeads Protein G (Invitrogen, Cat.# 10004D) were used per sample. After two washes with the native IP buffer, three washes were performed with a washing buffer (20 mM Tris pH 8.0, 125 mM NaCl, 0.5% (vol/vol) NP-40). Five biological replicates were performed per condition. Mass spectrometric analysis on a Q-Exactive HF Orbitrap (Thermo Scientific) and processing of the data was done as described previously[47].

## Reporting summary

Further information on research design is available in the Nature Portfolio Reporting Summary linked to this article.

# Data availability

HiS-NET-seq, standard NET-seq, HiChIP and ChIP-Rx data generated in this study have been deposited at the Gene Expression Omnibus (GEO) database under super series accession number GSE214594. The proteomics data generated in this study have been deposited in the ProteomeXchange Consortium via the PRIDE partner repository under accession code PXD04365. Publicly available datasets with the following accession numbers were used in this study: GSE158963 (SI-NET-seq K562 dTAG-BRD4, DMSO 2 h, dTAG7 2 h), GSE150625 (qPRO-seq K562 and PRO-seq K562), GSE60456 (GRO-cap K562), GSE158965 (ChIP-Rx K562 dTAG-BRD4 PAF1, DMSO 2 h, dTAG7 2 h), ENCSR109IQO (Total RNA-seq K562), ENCSR000CLW (RNA-seq NIH/3T3), ENCSR000AKP (ChIP-seq K562 H3K27ac), ENCSR000EWC (ChIP-seq K562 H3K4me1), ENCSR000EWA (ChIP-seq K562 H3K4me3), ENCSR000EWB (ChIP-seq K562 H3K27me3), ENCSR000AKR (ChIP-seq K562 H3K36me3), ENCSR000APD (ChIP-seq K562 H3K79me2). We used the human and mouse genome reference (GRCh38.p12; GRCm38.p6) and annotation (v28; M18) provided by Gencode. K562 enhancer were extracted from HACER. Source data are provided with this paper.

## Code availability

Computer code generated in this study is available at https://github.molgen.mpg.de/MayerGroup/HiS-NET-seq_paper_code.git.

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

## Acknowledgements

We thank Nicole Eischer and Mario Rubio for their critical comments on the manuscript. We thank Martyna Gajos for the discussions and Susanne Freier for cell culture support during the development of HiS-NET-seq. We thank Tuğçe Aktaş for advice with the enhancer assay. We thank Fiona Douglas for help with cloning of the enhancer constructs and the MPIMG Sequencing facility for sequencing. We thank David Meierhofer and the MPIMG Proteomics facility for MS measurements. Vector backbones for the enhancer assay were a generous gift from Daniel Ibrahim (Charité Berlin). This work was funded by the Max Planck Society (to A.M. and D.H.) and the Deutsche Forschungsgemeinschaft (DFG, grant no. 418415292 to A.M. and the International Research Training Group (IRTG) 2403 to A.M. and M.A.). O.J. was supported by a 2017 FEBS Long-Term Fellowship and D.H. by DFG grants HN 4/1-1, HN 4/3-1.

## Author contributions

A.B. and O.J. developed HiS-NET-seq. A.B. performed computational analyses of HiS-NET-seq, ChIP-Rx and HiChIP data. O.J. performed HiS-NET-seq experiments. M.A. performed HiS-NET-seq, ChIP-Rx, HiChIP and proteomics experiments. E.A. designed and established the enhancer reporter assay, and performed reporter assay experiments with the help of F.T. E.A. also performed HAT inhibitor and Western blot experiments. T.A.K. helped with the proteomics analysis. J.E.H. helped analyze HiS-NET-seq data. D.H. helped with HiChIP experiments. A.M. planned and designed experiments, and had overall responsibility over the study. A.M. wrote the manuscript with help of A.B., E.A., and M.A.

## Funding

## Competing interests

The authors declare no competing interests.
