## [Peer Review File · Nature Communications]

High-sensitive nascent transcript sequencing reveals BRD4-specific control of widespread enhancer and target gene transcriptionREVIEWER COMMENTS

Reviewer #1 (Remarks to the Author):

In this manuscript, Bressin et al. develop a new more sensitive NET-seq protocol called HiS-NET-seq, in which they use 4sU labeling to better purify native transcripts from recently transcribing Pol II. As such, the resulting libraries of the 3' ends represent the position of Pol II across the genome and nicely capture coding and non-coding/enhancer RNA, similar to GRO-seq/PRO-seq. They then revisit their previous finding of convergently transcribed antisense transcripts. They find that a large fraction may actually represent enhancer RNA, which is bidirectionally transcribed but one transcript was masked because it overlapped with the coding transcript. They further use their method to analyze what happens to the Pol II transcripts when Brd4 is rapidly depleted using spike-ins to accurately quantify the decay. They find that both the enhancer RNA and the coding RNA is similarly depleted at two time points, suggesting that they follow the same dynamics. To test whether Brd4-mediated transcription at enhancers and promoters is coordinated, they use H3K27ac HiChIP and report a reduction in genomic contacts.

Overall, the manuscript is built on a nice new method and has some interesting insights, but there is no overarching story that ties the pieces together. The main conclusion – that Brd4 coordinates enhancer and promoter transcription – is not sufficiently supported. While Brd4 depletion reduces transcripts at enhancers and promoters similarly over time, this is not evidence that they are physically linked. If Brd4 is independently required for transcription at enhancers and promoters, the observed results are exactly what one would expect. The H3K27ac HiChIP data are supposed to test the physical link more directly, but H3K27ac is likely itself reduced upon Brd4 depletion (Winter et al 2017), thus it is unclear whether the 3D organization is indeed altered in these cells or whether the assay now no longer detects these interactions efficiently. Given the wide spread of the focus of the paper and the fact that the role of Brd4 in condensate formation and 3D genome interactions is already well studied, the main message of the paper is not really justified.

However, I do see potential of the new method and feel that the manuscript could serve to clarify the differences between techniques (NET-seq, HiS-NET-seq, GRO-seq, PRO-seq and others) and shed more light onto enhancer transcription and the previously sometimes confusing annotations of antisense transcripts. In its current form, the manuscript does not strongly focus on this. While it is pointed out in Supplementary Table 1 that “Nuclear run-on methods [like GRO-seq] may fail to capture a subset of engaged RNA polymerases that can't resume transcription under the labeling conditions”, the authors do not use the opportunity to come to a firm conclusion on whether this hypothesis is correct. Although there are attempts in the Supplement to address this, the burning question for me remains: Is there an advantage of using HiT-NET-seq over GRO-seq/PRO-seq? The Supplement shows a correlation between these methods of almost 0.9, which indicates that they are essentially identical. However, HiS-NET-seq now includes 4sU labeling, thus it is possible that the new method also does not capture inactive Pol II, just like GRO-seq/PRO-seq. As far as I understand, GRO-seq/PRO-seq could theoretically still be different since these methods use sarkosyl to artificially induce Pol II elongation, while HiS-NET-seq is performed under conditions where Pol II naturally elongates (during 4sU labeling). Even if it turns out that the readout of HiT-NET-seq and GRO-seq/PRO-seq is essentially identical (not including potential differences in read counts and signal-to-noise ratio), this would still be an important result. The authors should not hide this result. They are already using the PRO-cap data to show the nature of the convergently transcribed antisense transcripts, thus they should not give the impression that HiT-NET-seq is the method of choice for mapping enhancer RNA. GRO-seq has already been successfully used for this purpose for many years. If HiS-NET-seq is better, we need to understand why.

Apart from the unclear message and wide focus, the manuscript could be improved in its writing and presentation. What was previously known and what is added by the current manuscript is often very vaguely described and as a result may seem lost (the difference between HiT-NET-seq and GRO-seq/PRO-seq) or inflated (e.g. intragenic enhancers are themselves not new, they just had not been associated with the previously observed CATs). The figures should ideally speak for themselves and should not require the intense back and forth between figure legend, main text, methods and supplement that I had to do to understand them. Some main figure panels are quite

trivial (e.g. Fig. 3b: there are quantitative differences in enhancer transcripts), while others that are interesting are hardly described (e.g. Fig. 5d: what I understand as Pol II pausing versus elongating Pol II). Many of the axes' labels or scales are quite cryptic, and there is unnecessary terminology (e.g. "ATU" for "antisense transcription unit" is not spelled out and could be replaced by "antisense transcripts"). I think if the authors could spell out their reasoning more clearly, this could be a much better paper.

Reviewer #2 (Remarks to the Author):

The manuscript provides insights into the coordination of transcription at enhancers and target genes in human cells. The authors developed an approach called HiS-NET-seq, which allows for sensitive detection of nascent transcripts at lowly transcribed regions, such as enhancers. HiS-NET-seq also used UMI and spike-in controls to reduce the reverse transcription artifacts. In addition, the authors used multi-omics approaches, including ChIP-Rx, IP-MS, and HiChIP, to investigate the involvement of BRD4 in the regulation of genome-wide enhancer transcription and its role in coordinating transcription at enhancers and target genes. The results of this study enhanced our understanding of the function of BRD4, which is linked to Pol II transcription at enhancers and target genes. Overall, this manuscript provides an insightful investigation into the role of the BRD4 in transcription at enhancers and target genes.

Major comments:

1. In Fig.2, the authors classified gene-associated Pol II antisense transcription into convergent antisense (CAT) and divergent antisense transcription (DAT). The authors provide a thoughtful biological discussion for CAT in L305-310, it will be nice to also provide a comparable discussion for DAT.

2. The findings presented in Fig. 5c demonstrate an increase in Pol II occupancy at the promoter-proximal region upon BRD4 loss. Any explanation for this observation?

Minor:

1. In Fig.1, the authors show that HiS-NET-seq has a higher sensitivity than NET-seq. It will be nice to demonstrate this with examples.

Reviewer #3 (Remarks to the Author):

Bressin et al have developed an improved NET-seq protocol, called high-sensitive native elongating transcript sequencing (HiS-NET-seq), which they use to identify thousands of transcribed enhancers that had not been previously detected in K562 cells. They show that this method recovers more unique reads and less unmapped reads than NET-seq and the related PRO-seq protocols, and therefore detects more active genes and transcribed enhancers. They then apply the technique to cells containing degradation-tagged BRD4, and detect many more transcriptional changes after rapid BRD4 depletion than previously reported. They find that promoter-proximal HiS-NET-seq signal increases, whereas gene body signal decreases after BRD4 depletion, suggesting that BRD4 is involved in Pol II pause-release. At enhancers there is also a, albeit less convincing, trend of decreased distal but increased proximal HiS-NET-seq signal, indicating that BRD4 has a role in Pol II elongation at enhancers as well. By combining BRD4 ChIP-seq with H3K27ac HiChIP in BRD4-depleted cells, Bressin et al investigate BRD4-dependent enhancer-promoter communication. They find both disrupted and enhanced contact frequencies after BRD4 depletion, where enhancer-promoter, enhancer-enhancer, and promoter-promoter interactions are decreased, whereas interactions between repressed, H3K27me3 marked, regions are increased. Finally, they perform BRD4 immunoprecipitation-mass spectrometry and identify interactors involved in chromatin configuration (Cohesin), as well as histone modifiers, Mediator and the PAF complex. Based on these observations, a model is proposed where BRD4 mediates genome

contacts to coordinate transcription elongation at enhancers and promoters.

This is a really impressive study with well-designed experiments, including spike-in normalizations, and the development of an improved nascent RNA sequencing technique. It is also very well written. It should be of wide interest to molecular biologists.

Minor comments

1. The claim that BRD4 coordinates transcription at enhancers and target genes is an over-statement. BRD4 could be controlling transcription at these regions separately. In fact, the change in HiS-NET-seq at regions where genomic interactions are disrupted after BRD4 depletion is very similar (although the difference is statistically significant) to regions in which chromatin contacts don't change (Fig 6e).
2. What could be the explanation for increased contact frequencies (at repressive regions) after BRD4 depletion (Fig 6a-e)? The number of enhanced contacts exceed disrupted contacts and surprisingly, transcription is decreased at these enhanced contacts (Fig 6e).
3. I'm not sure I understand Fig 6c. I believe that this shows the annotation of different genomic regions based on chromatin marks, and that these annotations are then used in Fig 6d. How do these HMM generated annotations differ from defining H3K4me1-enriched regions as enhancers and then dividing them into active and repressed based on H3K27ac and H3K27me3? Extragenic and intragenic location could also be defined based on current gene annotations. This part could be better explained in the text.
4. It would be good to discuss the limitation of the HiS-NET-seq method. For example that nascent RNA labeling with 4sU makes it difficult to measure transcription with this technique in tissues and organisms.

Point-by-point response to reviewers

We thank the reviewers for their thorough evaluation of our manuscript and their highly constructive comments. A detailed point-by-point response to all comments and suggestions of the reviewers are provided in this document. Based on the reviewer's critiques we have performed new experiments and computational analyses, and re-wrote parts of the manuscript.

Although the new experiments strengthen the HiChIP data and therefore also related conclusions, we softened our previous claim that BRD4 coordinates transcription at enhancers and associated genes based on comments from Reviewer 1 and Reviewer 3. We adapted the manuscript accordingly and now present the potential role of BRD4 in the coordination of enhancer and target gene transcription as an hypothesis. Furthermore, as suggested by Reviewer 1 we now put a stronger emphasize on the HiS-NET-seq approach as now already indicated by the new title of the revised manuscript. Importantly, we have performed additional comparative analysis between HiS-NET-seq, conventional NET-seq and PRO-seq data. From these analyses we conclude that HiS-NET-seq captures a greater fraction of paused/arrested Pol II in the promoter-proximal regions of genes as compared to PRO-seq. We now show parts of these results in main Figure 1 and discuss the observed differences between HiS-NET-seq and PRO-seq in greater depth (new paragraph in discussion section).

We now provide the tables (Supplementary Tables 2-4) containing the genomic coordinates of CATs, DATs and putative enhancer regions in BED format that can be used as annotation files for human K562 cells and can directly be loaded into the IGV genome browser. We hope that this will make the data even more accessible to colleagues with an interest in antisense and non-coding transcription. We believe that this will help to reduce the confusion about antisense transcription units. Finally, we further improved figures based on critiques raised by Reviewer 1.

During the revision, we detected and corrected a bug in our computational code identifying regions with H3K27ac and H3K4me1. This correction led to minor adjustments in the Figure panels related to putative enhancers (Fig. 3a - 3d, Fig. 4a - 4c, Fig. 4g/h, Fig. 5b/d/e, Supplementary Fig. 3, Supplementary Fig. 4a - 4d, Supplementary Fig. 4f, Supplementary Fig. 5c) and Fig. 2d/g. However, the corresponding results and conclusions did not change. We apologize for this.

We believe that the new data, the stronger focus on the HiS-NET-seq approach and clarifications that we added to the main text have significantly strengthened this work. Again, we thank the reviewers for their thoughtful assessment of our work. The full point-by-point response is provided below (blue letters).

REVIEWER COMMENTS

Reviewer #1 (Remarks to the Author):

In this manuscript, Bressin et al. develop a new more sensitive NET-seq protocol called HiS-NET-seq, in which they use 4sU labeling to better purify native transcripts from recently transcribing

Pol II. As such, the resulting libraries of the 3' ends represent the position of Pol II across the genome and nicely capture coding and non-coding/enhancer RNA, similar to GRO-seq/PRO-seq. They then revisit their previous finding of convergently transcribed antisense transcripts. They find that a large fraction may actually represent enhancer RNA, which is bidirectionally transcribed but one transcript was masked because it overlapped with the coding transcript. They further use their method to analyze what happens to the Pol II transcripts when Brd4 is rapidly depleted using spike-ins to accurately quantify the decay. They find that both the enhancer RNA and the coding RNA is similarly depleted at two time points, suggesting that they follow the same dynamics. To test whether Brd4-mediated transcription at enhancers and promoters is coordinated, they use H3K27ac HiChIP and report a reduction in genomic contacts.

Overall, the manuscript is built on a nice new method and has some interesting insights, but there is no overarching story that ties the pieces together. The main conclusion – that Brd4 coordinates enhancer and promoter transcription - is not sufficiently supported. While Brd4 depletion reduces transcripts at enhancers and promoters similarly over time, this is not evidence that they are physically linked. If Brd4 is independently required for transcription at enhancers and promoters, the observed results are exactly what one would expect. The H3K27ac HiChIP data are supposed to test the physical link more directly, but H3K27ac is likely itself reduced upon Brd4 depletion (Winter et al 2017), thus it is unclear whether the 3D organization is indeed altered in these cells or whether the assay now no longer detects these interactions efficiently.

We thank the reviewer for the thorough evaluation of our work and the thoughtful comments.

We agree with the reviewer that more research will be required to further clarify the links between transcription at enhancers and target genes in future. We have therefore softened the claim that BRD4 coordinates transcription at enhancers and cognate genes throughout the revised manuscript. We have also changed the title and the abstract of the manuscript accordingly. Although we still think that our findings support the view that transcription at enhancers and target genes is linked at least at a set of genes, we now present this model as an hypothesis and not as in the original manuscript as a conclusion.

We respectfully disagree with the reviewer's opinion that H3K27ac is likely reduced upon BRD4-specific depletion and the use of Winter et al., Mol Cell (2017) as a reference to support this view. In Winter et al., no experimental evidence is provided that pan-BET degradation or pan-BET inhibition impact H3K27Ac levels. In another study by Kedaigle et al. (Hum Mol Genet, 2020; PMID: 31696228) in which pan-BET inhibition in combination with H3K27Ac ChIP-seq was applied in murine embryonic stem cells, no global effects on H3K27Ac levels could be detected. Because of this and because of the short exposure with the degrader in our study (40 min and 2 h), we do not expect that acute BRD4-specific loss provokes a widespread reduction of H3K27Ac levels during the treatment time. However, we agree with the reviewer that a potential reduction in H3K27Ac upon acute BRD4-selective depletion would impact the HiChIP analyses. In order to clarify whether acute BRD4-specific loss reduces H3K27Ac levels in human K562 cells, we conducted new experiments. Notably, these experiments show that H3K27ac levels were not affected upon 2 h dTAG7 treatment whereas BRD4 levels were abolished. As expected, treatment with the histone acetyltransferase inhibitor A-485 (p300/CBP) caused a clear reduction of

H3K27Ac serving as a positive control. We added the new results that argue against an impact of BRD4-selective degradation on H3K27Ac levels during the short treatment with the BRD4-specific degrader dTAG7 to Supplementary Fig. 6 (New: Supplementary Fig. 6b; also shown below this response). This new data strengthens the HiChIP results and therefore also the evidence that BRD4 can link transcription at enhancers and their target genes through long-range regulatory contacts.

Given the wide spread of the focus of the paper and the fact that the role of Brd4 in condensate formation and 3D genome interactions is already well studied, the main message of the paper is not really justified.

We agree with the reviewer that BRD4 has been studied in the context of condensate formation and 3D genome regulation as we stated in the original manuscript. However, as we pointed out in the introduction and discussion sections, direct BRD4-specific roles in these processes have remained unclear mainly because pan-BET inhibition and pan-BET degradation have been used in the majority of prior functional studies. These methods cannot discriminate between individual BET protein functions. Furthermore, and as stated in the introduction section it is under debate whether BRD4 is implicated in 3D genome regulation. Although it was thought that BRD4 participates in higher-order genome organization and enhancer-promoter interactions, a recent study found that BET proteins are dispensable for these 3D genome contacts (Crump et al., Nat Commun, 2021; PMID: 33431820). We here provide evidence for a direct and BRD4-specific role in long-range 3D regulatory contacts between a set of enhancers and target genes. To the best of our knowledge we also provide for the first time evidence for a direct general role of BRD4 in the control of enhancer transcription. This was mainly accomplished by a combination of rapid BRD4-selective perturbation in human cells with approaches that capture the immediate consequences on transcription and 3D genome organization.

However, I do see potential of the new method and feel that the manuscript could serve to clarify the differences between techniques (NET-seq, HiS-NET-seq, GRO-seq, PRO-seq and others) and shed more light onto enhancer transcription and the previously sometimes confusing annotations of antisense transcripts. In its current form, the manuscript does not strongly focus on this. While it is pointed out in Supplementary Table 1 that “Nuclear run-on methods [like GRO-seq] may fail to capture a subset of engaged RNA polymerases that can’t resume transcription under the labeling conditions”, the authors do not use the opportunity to come to a firm conclusion on whether this hypothesis is correct. Although there are attempts in the Supplement to address this, the burning question for me remains: Is there an advantage of using HiT-NET-seq over GRO-seq/PRO-seq? The Supplement shows a correlation between these methods of almost 0.9, which indicates that they are essentially identical. However, HiS-NET-seq now includes 4sU labeling, thus it is possible that the new method also does not capture inactive Pol II, just like GRO-seq/PRO-seq. As far as I understand, GRO-seq/PRO-seq could theoretically still be different since these methods use sarkosyl to artificially induce Pol II elongation, while HiS-NET-seq is performed under conditions where Pol II naturally elongates (during 4sU labeling). Even if it turns out that the readout of HiT-NET-seq and GRO-seq/PRO-seq is essentially identical (not including potential differences in read counts and signal-to-noise ratio), this would still be an important result. The authors should not hide this result. They are already using the PRO-cap data to show the nature of the convergently transcribed antisense transcripts, thus they should not give the impression that HiT-NET-seq is the method of choice for mapping enhancer RNA. GRO-seq has already been successfully used for this purpose for many years. If HiS-NET-seq is better, we need to understand why.

We thank the reviewer for pointing this out. Based on this reviewer comment, we now put more emphasis on the HiS-NET-seq approach and its comparison with the other single-nucleotide RNA polymerase profiling methods in the revised manuscript. We have adapted the manuscript accordingly including the title that now indicates the new approach. Importantly, we have performed additional comparative analyses between HiS-NET-seq, conventional NET-seq and PRO-seq methods. The new analyses confirmed our observation that the main difference between the HiS-NET-seq and PRO-seq variants lies in the Pol II occupancy in the promoter-proximal region of genes. Interestingly, as compared to PRO-seq and qPRO-seq, HiS-NET-seq captures significantly more Pol II in the promoter-proximal region of genes (New Figure 1f; also shown below this response). Consistently, the correlation between the Pol II density in the promoter-proximal regions was lowest between HiS-NET-seq and PRO-seq (New Supplementary Fig. 2d; also shown below this response) and also lower as compared to the Pol II gene correlation between the different data sets (New Supplementary Fig. 2a; also shown below this response). These results suggest that 4sU labeling under natural transcription elongation conditions, as used in HiS-NET-seq, systematically captures Pol II complexes missed by the other labeling based methods. Interestingly, HiS-NET-seq consistently correlates best with conventional NET-seq, which captures arrested/paused Pol II. A potential explanation for this consense could be that 4sU labeling captures (HiS-NET-seq) likewise arrested/paused Pol II.

In addition, we found that HiS-NET-seq captures on average also more transcriptionally engaged Pol II that was closer to the TSS as compared to PRO-seq and qPRO-seq. Whereas the median peak occupancy of engaged Pol II in the promoter-proximal region was 80 nt downstream of the

TSS in HiS-NET-seq, it was 94 nt and 100 nt for PRO-seq and qPRO-seq, respectively (New Supplementary Fig. 2c; also shown below this response). Therefore, HiS-NET-seq potentially captures more immature Pol II complexes that localize closer to the TSS compared to other methods detecting exclusively RNA polymerases that are recovering from the pause further downstream from the TSS. However, we cannot rule out technical reasons for these observed differences during sequencing library preparation such as differences in the size selection of labeled nascent RNAs. The new data panels are also shown below this response.

Together, these findings support the view that HiS-NET-seq captures a greater fraction of the population of transcriptionally engaged RNA polymerases in the promoter-proximal region of genes including RNA polymerases that are paused or arrested. This represents an advantage over current PRO-seq methods. We now included parts of these findings in main Fig. 1 and added additional data panels to Supplementary Fig. 2. Furthermore, we now discuss the observed differences in more detail in a new paragraph in the discussion section. We think that this will further clarify the advantage of HiS-NET-seq over nuclear run-on based high-resolution profiling methods.

We agree with the reviewer that HiS-NET-seq sheds more light on enhancer transcription and antisense transcription units. We now provide the lists with the exact genomic coordinates of antisense transcription units (CATs and DATs; Supplementary Table 3) and enhancer regions (Supplementary Tables 2 and 4) as determined by HiS-NET-seq for human K562 cells in BED file format. The tables can be used as annotation files for CATs, DATs and enhancer regions, and can be directly loaded and displayed in the IGV genome browser. We think that this will help our colleagues to make better use of the large amount of HiS-NET-seq data that we have generated and of the integrative analyses that we have performed in this study. We hope that this will help to solve the confusion about antisense transcription units and transcribed enhancer regions at least in human K562 cells.

Apart from the unclear message and wide focus, the manuscript could be improved in its writing and presentation. What was previously known and what is added by the current manuscript is often very vaguely described and as a result may seem lost (the difference between HiT-NET-

seq and GRO-seq/PRO-seq) or inflated (e.g. intragenic enhancers are themselves not new, they just had not been associated with the previously observed CATs).

We added further clarification of what was known from previous studies and what we uncovered in this study to the main text of the revised manuscript. We now explicitly describe and discuss the observed main differences between HiS-NET-seq and PRO-seq methods (especially in the first results section and the discussion section). We also clarified that intragenic enhancers have been studied previously and added relevant references, but that HiS-NET-seq and the identification of CATs can now be used to systematically identify transcribed enhancers within active host genes (first paragraph of the discussion section).

The figures should ideally speak for themselves and should not require the intense back and forth between figure legend, main text, methods and supplement that I had to do to understand them. Some main figure panels are quite trivial (e.g. Fig. 3b: there are quantitative differences in enhancer transcripts), while others that are interesting are hardly described (e.g. Fig. 5d: what I understand as Pol II pausing versus elongating Pol II). Many of the axes' labels or scales are quite cryptic, and there is unnecessary terminology (e.g. "ATU" for "antisense transcription unit" is not spelled out and could be replaced by "antisense transcripts"). I think if the authors could spell out their reasoning more clearly, this could be a much better paper.

Based on this reviewer's critique, we have further improved figures. Specifically, we have clarified axes labels of Fig. 3c, Fig. 4c, Fig. 4f, Fig. 4g, Fig. 5c, Fig. 5d, Fig. 5f, Fig 6a, Fig. 6d and Supplementary Fig.1k, and eliminated unnecessary acronyms whenever possible. We also added additional clarification to Fig. 5d that visualizes the immediate impact on pausing and transcription elongation for two distinct time points upon acute loss of BRD4. For Fig. 6c we added an explanation to the main text. We believe that these additional clarifications will help future readers to immediately grasp the key points of the respective figure panel.

Reviewer #2 (Remarks to the Author):

The manuscript provides insights into the coordination of transcription at enhancers and target genes in human cells. The authors developed an approach called HiS-NET-seq, which allows for sensitive detection of nascent transcripts at lowly transcribed regions, such as enhancers. HiS-NET-seq also used UMI and spike-in controls to reduce the reverse transcription artifacts. In addition, the authors used multi-omics approaches, including ChIP-Rx, IP-MS, and HiChIP, to investigate the involvement of BRD4 in the regulation of genome-wide enhancer transcription and its role in coordinating transcription at enhancers and target genes. The results of this study enhanced our understanding of the function of BRD4, which is linked to Pol II transcription at enhancers and target genes. Overall, this manuscript provides an insightful investigation into the role of the BRD4 in transcription at enhancers and target genes.

We thank this reviewer for the highly constructive and helpful comments.

Major comments:

1. In Fig.2, the authors classified gene-associated Pol II antisense transcription into convergent antisense (CAT) and divergent antisense transcription (DAT). The authors provide a thoughtful biological discussion for CAT in L305-310, it will be nice to also provide a comparable discussion for DAT.

Thank you for pointing this out. We now also included a more detailed discussion for divergent antisense transcription (DAT) in the revised manuscript. Based on this reviewer comment we also performed a new comparative analysis of DAT and CAT locations in order to identify potential distinctive features. This new analysis revealed that H3K4me3 is significantly enriched at DAT sites, representing a main difference between both types of antisense transcription in human cells (New: Supplementary Figure 2f; also shown below this response). We added this new finding to the second results paragraph that is related to Figure 2. However, we would like to keep the overall focus on CAT units because they turned out to be indicators for intragenic putative enhancers which is a main theme of the present study.

2. The findings presented in Fig. 5c demonstrate an increase in Pol II occupancy at the promoter-proximal region upon BRD4 loss. Any explanation for this observation?

We apologize for the insufficient description of the findings shown in Fig. 5c. As an immediate consequence of acute BRD4-specific degradation, HiS-NET-seq revealed an accumulation of Pol II in the promoter-proximal region of genes that was accompanied by a decrease of Pol II occupancy along the gene-body (Fig. 5c). These observations are indicative for a pause-release defect of Pol II and were consistent with results of our previous study (*Arnold, *Bressin et al., Mol Cell, 2021; PMID: 34324863). With the greater sensitivity of HiS-NET-seq as compared to conventional NET-seq, we now observed a pause-release defect at nearly all active genes upon acute loss of BRD4 providing a more complete view of the effect. We added clarification to the main text.

Minor:

1. In Fig.1, the authors show that HiS-NET-seq has a higher sensitivity than NET-seq. It will be nice to demonstrate this with examples.

We thank the reviewer for this comment. We now included two representative gene examples (New Supplementary Fig. 1j; also shown below this response) to further illustrate the higher sensitivity of HiS-NET-seq over conventional NET-seq in detecting transcriptionally engaged RNA polymerase along genes. The new Figure panel is also shown below.

Reviewer #3 (Remarks to the Author):

Bressin et al have developed an improved NET-seq protocol, called high-sensitive native elongating transcript sequencing (HiS-NET-seq), which they use to identify thousands of transcribed enhancers that had not been previously detected in K562 cells. They show that this method recovers more unique reads and less unmapped reads than NET-seq and the related PRO-seq protocols, and therefore detects more active genes and transcribed enhancers. They then apply the technique to cells containing degradation-tagged BRD4, and detect many more transcriptional changes after rapid BRD4 depletion than previously reported. They find that promoter-proximal HiS-NET-seq signal increases, whereas gene body signal decreases after BRD4 depletion, suggesting that BRD4 is involved in Pol II pause-release. At enhancers there is also a, albeit less convincing, trend of decreased distal but increased proximal HiS-NET-seq signal, indicating that BRD4 has a role in Pol II elongation at enhancers as well. By combining BRD4 ChIP-seq with H3K27ac HiChIP in BRD4-depleted cells, Bressin et al investigate BRD4-dependent enhancer-promoter communication. They find both disrupted and enhanced contact frequencies after BRD4 depletion, where enhancer-promoter, enhancer-enhancer, and promoter-promoter interactions are decreased, whereas interactions between repressed, H3K27me3 marked regions are increased. Finally, they perform BRD4 immunoprecipitation-mass

spectrometry and identify interactors involved in chromatin configuration (Cohesin), as well as histone modifiers, Mediator and the PAF complex. Based on these observations, a model is proposed where BRD4 mediates genome contacts to coordinate transcription elongation at enhancers and promoters.

This is a really impressive study with well-designed experiments, including spike-in normalizations, and the development of an improved nascent RNA sequencing technique. It is also very well written. It should be of wide interest to molecular biologists.

We thank this reviewer for the highly constructive and thoughtful comments, and the encouraging words.

Minor comments

1. The claim that BRD4 coordinates transcription at enhancers and target genes is an overstatement. BRD4 could be controlling transcription at these regions separately. In fact, the change in HiS-NET-seq at regions where genomic interactions are disrupted after BRD4 depletion is very similar (although the difference is statistically significant) to regions in which chromatin contacts don't change (Fig 6e).

We thank the reviewer for pointing this out. We have softened the claim that BRD4 coordinates transcription at enhancers and target genes throughout the revised manuscript. We have also changed the title and the abstract of the manuscript accordingly. Although, we still think that our findings support the view that transcription at enhancers and target genes is coordinated at least at a set of genes, we now present this model as an hypothesis. We agree with the reviewer that more research will be required to clarify the coordinated regulation of transcription in future. We added a statement to clarify this.

2. What could be the explanation for increased contact frequencies (at repressive regions) after BRD4 depletion (Fig 6a-e)? The number of enhanced contacts exceed disrupted contacts and surprisingly, transcription is decreased at these enhanced contacts (Fig 6e).

We thank the reviewer for this thoughtful question. As shown in Fig. 6d, the average interaction frequencies between most *regulatory contacts* -contacts that include regulatory elements 'promoters' and 'enhancers'- immediately decreased after BRD4-specific degradation. However, in the statistical test (Fig. 6a), indeed a set of interactions emerged as significantly increased. 'Enhanced contacts' were mainly detected between 'repressed' and 'low signal' chromatin regions (Fig. 6d). Consistently, 'enhanced contacts' had no specific link to histone marks associated with regulatory elements (Fig. 6b). We therefore speculate that these changes are rather unspecific secondary effects that may arise from the massive deregulation of Pol II transcription in the absence of BRD4. In our previous study (*Arnold, *Bressin et al., Mol Cell, 2021; PMID: 34324863), we found that BRD4 loss caused massive readthrough transcription that can span hundreds of kilobases downstream of the last active polyA site of genes. Transcription at gene ends and at downstream repressed regions likely open the chromatin and may cause random

interactions between usually 'repressed' and 'low-signal' genomic sites now detected as enhanced contact frequencies by HiChIP upon BRD4 loss. Our observation that 'enhanced contacts' were over-represented mainly among the contact type classes 'Gene-body - Gene-body', 'Repressed - Repressed' and '3'-end - 3'-end' (Supplementary Fig. 6c) further supports this view.

3. I'm not sure I understand Fig 6c. I believe that this shows the annotation of different genomic regions based on chromatin marks, and that these annotations are then used in Fig 6d. How do these HMM generated annotations differ from defining H3K4me1-enriched regions as enhancers and then dividing them into active and repressed based on H3K27ac and H3K27me3? Extragenic and intragenic location could also be defined based on current gene annotations. This part could be better explained in the text.

We thank the reviewer for this comment. We now added an explanation of Fig. 6c to the main text of the revised manuscript. As correctly described by the reviewer, Fig. 6c shows the genome annotation using chromatin marks that allow the interpretation of observed 3D contact changes as measured by HiChIP at a ten-kilobase resolution. We have chosen this annotation strategy, provided by chromHMM, because it allows a genome-wide annotation at a desired resolution (here: ten kilobases) that fits our HiChIP results. In contrast to alternative annotation strategies, the underlying Hidden-Markov Model identifies an annotation state for each genomic window, considering an optimized mathematical model. Therefore, this approach is less biased as a proposed rule-based assignment, which is particularly important if multiple gene regulatory features occur within the same ten-kilobase bin.

4. It would be good to discuss the limitation of the HiS-NET-seq method. For example that nascent RNA labeling with 4sU makes it difficult to measure transcription with this technique in tissues and organisms.

Thank you for pointing this out. We now discuss current limitations of the HiS-NET-seq approach in the discussion section of the revised manuscript. We have already started to experimentally address some of these limitations but this will be beyond this study.

REVIEWERS' COMMENTS

Reviewer #1 (Remarks to the Author):

In the revised version, Bressin et al. tell a much more convincing story. They start with describing HiS-NET-seq, its sensitivity in detecting nascent enhancer transcripts, including those found in transcribed regions, and how it compares to other state-of-the-art techniques such as PRO-seq. They then use HiS-NET-seq to analyze the response to acute BRD4 depletion. They find that nascent transcripts are rapidly depleted (at 40 min and 2h time points) at both enhancer and promoter regions where BRD4 is also bound, suggesting that BRD4 regulates transcript elongation at enhancers at promoters. In addition, HiChIP analysis (using H3K27ac) and immunoprecipitation followed by mass spec provide evidence that BRD4 participates in long-range interactions between enhancers and promoters.

Overall, the manuscript is much improved and now presents convincing evidence that HiS-NET-seq is useful and that BRD4 is required for enhancer transcription similar to its role at promoters. The authors have also attempted to make the figures more understandable, which I appreciate, but there is still room for improvement, especially when it comes to the motivation, fluency and logical flow of the writing. This manuscript could still be much better, especially given the writing standards that LLMs now easily provide, but I leave this to the editors to handle.

Major point:

The authors deposited the raw data in GEO but no processed data or code is provided. For this manuscript, "available upon request" is unacceptable. Many of the claims rely on the better identification of enhancer transcripts so the analysis and discovered enhancer transcript coordinates are key.

Minor points:

The title now represents the main finding of the paper well, but only after reading the paper carefully. On its own, it is very long and it is not clear what "direct" in the title means. It certainly does not reflect a direct role in transcription. If "direct" is important, I would focus on the widespread role in enhancer transcription. Or one could leave out "direct" and "widespread", and focus on the "enhancer and target gene transcription".

The overall motivation of the study is not well described. "A complete picture of the Pol II transcriptional landscape has not yet emerged" is not a good argument. When will the picture ever be complete? How do we know that we are missing key information? What is the overarching goal of understanding transcription? In my view, what's missing is something along the lines of the long-term goal is to decode non-coding sequence information, which is facilitated by mapping enhancer activity at the highest resolution and sensitivity, and understanding how enhancer activity leads to target gene activation.

Likewise, the motivation for studying BRD4 is not clear, BET family proteins in general and BRD4 specifically. To my knowledge, BET inhibitors are of interest from the therapeutic side, but that's not described. If one would like to understand enhancer activation and interactions with the promoter, why not study Mediator? Explaining what makes Brd4 so appealing is important for underscoring the significance of the findings.

It is not clear why the term "trans-acting factors" is used. It is used in reference to a hypothesis paper (TAG model), but is commonly not referred to in this way. Trans usually refers to proteins not acting in cis and thus can affect many genes. Isn't connecting enhancers to promoters strictly speaking still in cis? In any case, I would not use terms that are unclear without defining them properly. It would however help to introduce the open question of how enhancers communicate with enhancers a bit more since this is one of the key motivations for the study.

The words "multi-omics analysis" (abstract, main text) and "using Integrative functional multi-omics" are not the right terms. I associate "multi-omics" with assays that measure multiple

genomics modalities at once, ideally as a single-cell assay. Simply making inferences by comparing different genomics data sets with each other is a "genomics analysis".

Not sure the use of "ChIP-Rx" as a ChIP-seq experiment with spike-in control is necessary. The concept of a spike-in is much older than the cited references. The first reference I can think of is van de Peppel et al. EMBO Rep (2003) "Monitoring global messenger RNA changes in externally controlled microarray experiments". It has also been done for ChIP experiments without giving it a new name, so it's not clear to me why the term is used here. If it simply refers to the exact protocol, it can be cited in the methods.

In Figure 4f-h is a very lengthy and uninformative way of showing that HiS-NET-seq results in more called peaks than NET-seq. While I can see the differences visually, there is no mention of the method (DEseq2 according to the methods) and the numbers have no meaning as far as I can tell (similar to Figure 5c,f). It's not clear what panel h adds to this and why the numbers are different here. Another source of confusion is that the signal of HiS-NET-seq is sometimes referred to as transcription (main text) and sometimes as Pol II occupancy (in the figure here). Either one is fine, but it should be consistent and clearly introduced and labeled in the figures.

In Figure 5a, it is not clear what the + track (purple) and the - track (red) represents. I assume it is the strand, but this should be clearly labeled and described since it could also refer to with treatment (+) and without treatment (-).

In the methods, it says that Pol II density right at the TSSs and polyadenylation sites were masked. Why?

Reviewer #2 (Remarks to the Author):

The authors have successfully addressed all of my concerns.

Reviewer #3 (Remarks to the Author):

All of my comments have been satisfactorily addressed, and I find the revised manuscript to be an improved version that should be published.

Point-by-point response to reviewer 1:

Reviewer 1 raised several new comments at this stage of the revision. A detailed point-by-point response to all comments of this reviewer is provided in this document. Based on the reviewer's critique, we added additional clarifications to the revised manuscript and made the computer code that we generated in this study available at our Github site. The full point-by-point response is provided below (blue letters).

Reviewer 1:

In the revised version, Bressin et al. tell a much more convincing story. They start with describing HiS-NET-seq, its sensitivity in detecting nascent enhancer transcripts, including those found in transcribed regions, and how it compares to other state-of-the-art techniques such as PRO-seq. They then use HiS-NET-seq to analyze the response to acute BRD4 depletion. They find that nascent transcripts are rapidly depleted (at 40 min and 2h time points) at both enhancer and promoter regions where BRD4 is also bound, suggesting that BRD4 regulates transcript elongation at enhancers at promoters. In addition, HiChIP analysis (using H3K27ac) and immunoprecipitation followed by mass spec provide evidence that BRD4 participates in long-range interactions between enhancers and promoters.

Overall, the manuscript is much improved and now presents convincing evidence that HiS-NET-seq is useful and that BRD4 is required for enhancer transcription similar to its role at promoters. The authors have also attempted to make the figures more understandable, which I appreciate, but there is still room for improvement, especially when it comes to the motivation, fluency and logical flow of the writing. This manuscript could still be much better, especially given the writing standards that LLMs now easily provide, but I leave this to the editors to handle.

Major point:

The authors deposited the raw data in GEO but no processed data or code is provided. For this manuscript, "available upon request" is unacceptable. Many of the claims rely on the better identification of enhancer transcripts so the analysis and discovered enhancer transcript coordinates are key.

The statement of this reviewer that 'no processed data or code is provided' is incorrect. With the GEO accession number, we provide access to all NGS-associated data, including HiS-NET-seq, standard NET-seq, ChIP-Rx and HiChIP data. For all datasets, we provide raw (FASTQ) and the corresponding processed data files, including:

- Pol II occupancy tracks (+ and -) as bigwig files for HiS-NET-seq and standard NET-seq data.
- Input normalized bigwig files for ChIP-Rx data.
- Contact matrix at a 10 kb resolution for HiChIP data.

Furthermore, we provided source code access for the reviewers as requested during the review process. However, we agree that the source code should also be accessible to our colleagues. Therefore, we now added a reference to the public Github repository of our lab containing the complete source code for reproducing our study.

Minor points:

The title now represents the main finding of the paper well, but only after reading the paper carefully. On its own, it is very long and it is not clear what “direct” in the title means. It certainly does not reflect a direct role in transcription. If “direct” is important, I would focus on the widespread role in enhancer transcription. Or one could leave out “direct” and “widespread”, and focus on the “enhancer and target gene transcription”.

We removed 'direct' from the title of the manuscript.

The overall motivation of the study is not well described. “A complete picture of the Pol II transcriptional landscape has not yet emerged” is not a good argument. When will the picture ever be complete? How do we know that we are missing key information? What is the overarching goal of understanding transcription? In my view, what's missing is something along the lines of the long-term goal is to decode non-coding sequence information, which is facilitated by mapping enhancer activity at the highest resolution and sensitivity, and understanding how enhancer activity leads to target gene activation.

We disagree with this opinion. The different types of motivation for this study are indicated at the end of each paragraph of the introduction section. We further clarify the motivation for the different analyses that we have performed in this work at the beginning of each results section. We now further specified the motivation statement that was picked by the reviewer.

Likewise, the motivation for studying BRD4 is not clear, BET family proteins in general and BRD4 specifically. To my knowledge, BET inhibitors are of interest from the therapeutic side, but that's not described. If one would like to understand enhancer activation and interactions with the promoter, why not study Mediator? Explaining what makes Brd4 so appealing is important for underscoring the significance of the findings.

We disagree. The rationale for studying the BET family protein BRD4 is described in paragraph 4 of the introduction section and again at the beginning of the 4th paragraph of the results section entitled 'BRD4 regulates enhancer transcription genome-wide'. Although the disease relevance of BRD4 was explicitly mentioned in all previous versions of our manuscript, we now added an additional statement to the introduction to emphasize that BRD4 has also emerged as a therapeutic target.

It is not clear why the term “trans-acting factors” is used. It is used in reference to a hypothesis paper (TAG model), but is commonly not referred to in this way. Trans usually refers to proteins not acting in cis and thus can affect many genes. Isn't connecting enhancers to promoters strictly speaking still in cis? In any case, I would not use terms that are unclear without defining them properly. It would however help to introduce the open question of how enhancers communicate with enhancers a bit more since this is one of the key motivations for the study.

We replaced 'trans-acting factors' by 'transcription factors'. We now added an additional statement to emphasize that our knowledge of the molecular mechanisms of enhancer-target gene communication in the nucleus of cells is still incomplete.

The words “multi-omics analysis” (abstract, main text) and “using Integrative functional multi-omics” are not the right terms. I associate “multi-omics” with assays that measure multiple genomics modalities at once, ideally as a single-cell assay. Simply making inferences by comparing different genomics data sets with each other is a “genomics analysis”.

We disagree and do not share the reviewer’s opinion that the term ‘multi-omics’ refers to ‘assays that measure multiple genomics modalities at once’. The term is widely used to describe the combined usage of different ‘omics’ methods as we do in this multi-omics study. In the present work, we combined state-of-the-art genome-wide approaches, also known as genomics methods, transcriptome-wide approaches, also known as transcriptomics methods, and proteome-wide approaches, also known as proteomics methods, as we described in detail in the methods section.

Not sure the use of “ChIP-Rx” as a ChIP-seq experiment with spike-in control is necessary. The concept of a spike-in is much older than the cited references. The first reference I can think of is van de Peppel et al. EMBO Rep (2003) “Monitoring global messenger RNA changes in externally controlled microarray experiments”. It has also been done for ChIP experiments without giving it a new name, so it’s not clear to me why the term is used here. If it simply refers to the exact protocol, it can be cited in the methods.

The precise term of the spike-in controlled ChIP-seq method that we have used in this study is ‘ChIP with reference exogenous genome’ or short ‘ChIP-Rx’ (PMID: 25437568). We think that this term is necessary to indicate which protocol we have used in this work and also to give credit to the original inventors of this powerful and widely used ChIP protocol. We are aware that the concept of spike-ins is older than that.

In Figure 4f-h is a very lengthy and uninformative way of showing that HiS-NET-seq results in more called peaks than NET-seq. While I can see the differences visually, there is no mention of the method (DEseq2 according to the methods) and the numbers have no meaning as far as I can tell (similar to Figure 5c,f).

It is not clear to us what this reviewer means by her/his statement ‘that HiS-NET-seq results in more called peaks than NET-seq’, since the corresponding figures do not depict peak calling results. We choose the volcano plot for visualizing the significant changes in Pol II occupancy at specified regions across Pol II profiling methods (conventional NET-seq and HiS-NET-seq) after BRD4 degradation as described in the corresponding figure legends. Both figures emphasize the importance of using the high-sensitivity NET-seq approach when studying gene or enhancer Pol II occupancy changes. As depicted, the obtained results may differ substantially.

As correctly stated by the reviewer, we reference in the figure legend the corresponding methods section, which describes the DEseq2-based approach that we have used. We state the meaning of the numbers in all corresponding figure legends, too. For Figures 4f, 4g and 5c, we refer to either genes or enhancer regions with significantly changed Pol II occupancy. In Fig. 5f, we highlight BRD4 binding sites that change significantly. We now report the applied method in the respective figure legends as requested by the reviewer.

It's not clear what panel h adds to this and why the numbers are different here.

Fig. 4h integrates -in contrast to Fig. 4g- results from our HiS-NET-seq and BRD4 ChIP-Rx analyses. In the main text and the figure legend, we define BRD4-sensitive enhancers as a subset of putative enhancers that show a significant reduction in DNA-bound BRD4 after BRD4 degradation using ChIP-Rx. Interestingly, and as shown in Fig. 4h, the strongest reduction of Pol II occupancy levels occur at those BRD4-sensitive putative enhancers.

Another source of confusion is that the signal of HiS-NET-seq is sometimes referred to as transcription (main text) and sometimes as Pol II occupancy (in the figure here). Either one is fine, but it should be consistent and clearly introduced and labeled in the figures.

As described in the main text of the manuscript, HiS-NET-seq captures RNA polymerase that is engaged in transcription with nucleotide precision. Therefore, we think that the terms 'Pol II occupancy', 'transcription' and 'nascent transcription' can be used interchangeably. We do not share the reviewer's opinion that this creates 'another source of confusion'.

In Figure 5a, it is not clear what the + track (purple) and the - track (red) represents. I assume it is the strand, but this should be clearly labeled and described since it could also refer to with treatment (+) and without treatment (-).

(+) and (-) refer to DNA strands. We choose this widely used DNA strand labeling scheme over the 'sense/antisense' scheme to avoid confusion. The 'antisense transcription units' as we defined them in this study are not exclusively located on the (-) strand. We use this labeling scheme consistently in all figure panels of the manuscript that show Pol II occupancy tracks, and hence not only in Fig. 5a.

In the methods, it says that Pol II density right at the TSSs and polyadenylation sites were masked. Why?

As stated in the methods section entitled '*Processing of NET-seq and HiS-NET-seq data*', sequencing reads that originated from RNA processing intermediates, mainly generated during RNA splicing and 3'-end RNA cleavage, were masked during NET-seq data analysis. We excluded sequencing reads mapping to the 3'-most nucleotide position of annotated intron and exon boundaries, including the polyadenylation site. This results in an artificial drop of Pol II occupancy at transcription start and polyadenylation sites in metagene profiles. Subsequently, we mask these nucleotide positions in metagene visualizations to avoid any misinterpretation of the data. We added an additional clarification to the corresponding methods section of the manuscript.

Reviewer 2:

The authors have successfully addressed all of my concerns.

Reviewer 3:

All of my comments have been satisfactorily addressed, and I find the revised manuscript to be an improved version that should be published.